# Hierarchical Agglomerative Graph Clustering in Poly-Logarithmic Depth

**Laxman Dhulipala**
University of Maryland and
Google Research
laxman@umd.edu

**David Eisenstat**
Google Research
eisen@google.com

**Jakub Łącki**
Google Research
jlacki@google.com

**Vahab Mirrokni**
Google Research
mirrokni@google.com

**Jessica Shi**
MIT CSAIL
jeshi@mit.edu

## Abstract

Obtaining scalable algorithms for *hierarchical agglomerative clustering* (HAC) is of significant interest due to the massive size of real-world datasets. At the same time, efficiently parallelizing HAC is difficult due to the seemingly sequential nature of the algorithm. In this paper, we address this issue and present ParHAC, the first efficient parallel HAC algorithm with sublinear depth for the widely-used average-linkage function. In particular, we provide a $(1 + \epsilon)$-approximation algorithm for this problem on $m$ edge graphs using $\tilde{O}(m)$ work and poly-logarithmic depth. Moreover, we show that obtaining similar bounds for *exact* average-linkage HAC is not possible under standard complexity-theoretic assumptions.

We complement our theoretical results with a comprehensive study of the ParHAC algorithm in terms of its scalability, performance, and quality, and compare with several state-of-the-art sequential and parallel baselines. On a broad set of large publicly-available real-world datasets, we find that ParHAC obtains a 50.1x speedup on average over the best sequential baseline, while achieving quality similar to the exact HAC algorithm. We also show that ParHAC can cluster one of the largest publicly available graph datasets with 124 billion edges in a little over three hours using a commodity multicore machine.

## 1  Introduction

Hierarchical Agglomerative Clustering (HAC) [49, 51, 72] is a fundamental and widely-used clustering method with numerous applications in unsupervised learning, community detection, and biology. Given $n$ input points, the HAC algorithm starts by forming a separate cluster for each input point, and proceeds in $n - 1$ steps. Each step replaces the two most *similar* clusters by its union. The exact notion of similarity between two clusters is specified by a configurable *linkage function*. This function is typically given all pairwise similarities between points from the two clusters. Some of the most popular choices are *average-linkage* (the arithmetic mean of all similarities), *single-linkage* (the maximum similarity), and *complete-linkage* (the minimum similarity). Among these, average-linkage is of particular importance, as it is known to find high-quality clusters in real-world applications [25, 55, 56, 65].[1]

---

[1]We note that HAC can also be defined in terms of input points and *dissimilarities* between the points. We discuss and compare both settings in Section 1.2

36th Conference on Neural Information Processing Systems (NeurIPS 2022).

Although HAC has been of significant interest to statisticians, computer scientists, and clustering practitioners since the 1960s, applying HAC to very large datasets remains a major challenge. A significant source of difficulty is the need to compute all pairwise similarities between sets of points, which HAC implementations typically perform at the start of the algorithm at the expense of $\Theta(n^2)$ work and space [58, 59, 76]. To address this difficulty, two directions have recently been explored in the literature. The first has focused on designing approximate sub-quadratic work algorithms in the dissimilarity setting, using sketching and approximate nearest neighbor (ANN) techniques [4, 57]. The second approach focuses on the *similarity-graph setting*, and is what we build on in this paper. Here, the idea is to build a (typically sparse) similarity graph over the pointset input, e.g., by representing each point as a vertex, connecting a point to its $k$ most similar neighbors, and then applying a graph-based HAC algorithm on this similarity graph [38].[2] Surprisingly, despite only using a sub-quadratic number of similarities, similarity graph-based HAC algorithms can match or sometimes even surpass the quality of an algorithm using the full $O(n^2)$ similarity matrix [38].

Prior works on these approaches [4, 38] give provable guarantees on the quality of the resulting approximation algorithms; in fact, both yield $(1 + \epsilon)$-approximate HAC algorithms, as proposed by [57]. Specifically, a $(1 + \epsilon)$-*approximate HAC algorithm* is an algorithm in which each step only merges edges that have similarity at least $\mathcal{W}_{\max}/(1 + \epsilon)$ where $\mathcal{W}_{\max}$ is currently the largest similarity.[3] Hence, an $(1 + \epsilon)$-approximate algorithm is constrained to merge an edge that is "close" in similarity to the merge that the exact algorithm would perform. At the same time, the algorithm has flexibility in which edge to merge, which can potentially be algorithmically exploited.

Despite the progress on scaling up HAC, further improvements appeared challenging due to the seemingly inherently sequential nature of the HAC algorithm [8, 55]. Several recent attempts at parallelizing HAC either rely on optimistic assumptions on the input [73, 79], significantly diverge from the original algorithm, potentially impacting quality [8, 23, 55], or deliver weak approximation guarantees [57]. Understanding if HAC can be solved in sub-quadratic work and poly-logarithmic depth (or even polynomial work and sub-linear depth) has thus remained an intriguing open question, even when allowing for approximation.

## 1.1 Our Contributions

In this paper, we introduce the first efficient parallel algorithm computing $(1 + \epsilon)$-approximate average-linkage HAC with poly-logarithmic depth and near-linear total work. We use the standard *work-depth model* of parallel computation [14, 26] to analyze the theoretical cost of our algorithm, where briefly, the *work* is the total number of operations performed, and the *depth* is the longest chain of dependencies. The work-depth model is the parallel model used in a variety of recent papers on shared-memory multicore algorithms for graph problems [31–33, 35, 71] pointset clustering [79], as well as tree-based algorithms [74, 75]. Deriving good bounds for the work and depth of an algorithm also implies good bounds for the problem in a variety of other models for parallelism, such as different PRAM variants, by using efficient simulation results (e.g., see [14, 15]).

The algorithm takes as input a similarity graph, which contains $n$ vertices (representing input points) and $m$ weighted edges (representing nonzero similarities between points).

**Theorem 1.1.** *There is a parallel $(1 + \epsilon)$-approximate average-linkage graph-based HAC algorithm that given a similarity graph containing $n$ vertices and $m$ weighted edges runs in $\tilde{O}(m + n)$ work in expectation and has $O(\log^4 n)$ depth with high probability.[4]*

For a fixed $\epsilon$, our algorithm, which we call ParHAC, runs in poly-logarithmic depth and $\tilde{O}(m)$-work. Thus, it is work-efficient up to logarithmic factors and achieves high parallelism (ratio of work to depth). We note that the same setting, i.e., parametrizing the input size by the number of *nonzero* similarities, has been recently studied in the sequential case [38], where an $\tilde{O}(m)$-work sequential $(1 + \epsilon)$-approximate algorithm is known. Prior to our result, no average-linkage HAC algorithm with sublinear depth was known (even allowing approximation).

---

[2]We note that [38] introduce both an exact algorithm for average-linkage HAC running in $O(n\sqrt{m})$ work and an approximate algorithm running in $\tilde{O}(m)$ work. We define $\tilde{O}(f(x)) := O(f(x)\mathsf{polylog}(f(x)))$.

[3]We note that [57] deals with dissimilarities, but we adapt the definition to similarities in the natural way.

[4]An algorithm has $O(f(n))$ cost with high probability (*whp*) if it has $O(k \cdot f(n))$ cost with probability at least $1 - 1/n^k$.

On the negative side, we show that allowing approximation is necessary, as obtaining an average-linkage *exact* HAC algorithm with poly-logarithmic depth is not possible under standard complexity-theory assumptions (i.e., it is a P-complete problem). In other words, an exact polynomial-work parallel HAC algorithm for average-linkage in poly-logarithmic depth would imply poly-logarithmic depth algorithms for *all* problems solvable in polynomial time (i.e., show that P = NC). Our lower bound formalizes a commonly held belief that the algorithm is inherently sequential [73] and justifies studying the approximate variant of HAC.

**Theorem 1.2.** *Graph-based HAC using average-linkage is* P-*complete.*

We complement our theoretical results by providing an efficient parallel implementation of ParHAC, and comparing it with existing graph-based and pointset-based HAC baselines across a broad set of publicly available real-world datasets. For scalability, we show that ParHAC achieves strong speedups relative to other high-quality HAC baselines, achieving 50.1x average speedup over the approximate sequential algorithm from [38] when run on a 72-core machine. Moreover, we study the overall running times of using ParHAC in the pointset setting, and find that ParHAC can cluster significantly larger datasets compared to fastcluster [59], a state-of-the-art pointset clustering algorithm, and achieves up to 417x speedup over fastcluster in end-to-end running time.

We show that the above speedups are achieved without loss (or even with gain) for the final algorithm. In particular, for the Adjusted Rand-Index quality measure, we find that the quality of ParHAC is on average 3.1% better than the best approximate sequential baseline, and is on average within 3.69% of the best score for any method on the datasets that we study. Finally, we find that ParHAC achieves consistently strong quality results on three other quality metrics that we study compared with the best solutions offered by exact and approximate algorithms.

We have made our implementation publicly available on GitHub.[5]

## 1.2  Graph-Based Hierarchical Agglomerative Clustering

As mentioned earlier in the introduction, our focus in this paper is on clustering (typically sparse) weighted graphs where the edge weights represent similarities between vertices. More formally, we study parallel algorithms for the *graph-based hierarchical agglomerative clustering (HAC)* problem, which takes as input a graph $G = (V, E, w)$ and proceeds by repeatedly merging the two most similar clusters, where the similarity is given by a configurable linkage function. Equivalently, this process can be viewed in terms of a graph $H$, whose vertices are clusters, and edges represent clusters of positive similarity (in particular, the edge weights give the similarities between clusters). Initially, when all clusters have size 1, $H$ is equal to $G$. Each operation of merging two clusters corresponds to contracting an edge of $H$ (we call this operation a *merge*). In the following, we typically use this alternative view of the algorithm.

The output of the algorithm is a *dendrogram* – a rooted binary tree that has a single leaf for each vertex in the input graph, with internal nodes corresponding to clusters obtained by merges, and weights of internal nodes representing the similarity of the corresponding merge that formed it.

We focus on the *average-linkage measure*, which assumes that the similarity between two clusters $(X, Y)$ is equal to $\sum_{(x,y)\in\mathsf{Cut}(X,Y)} w(x, y)/(|X| \cdot |Y|)$, that is, the total weight of edges between $X$ and $Y$, divided by the maximum number of possible edges between the clusters. Throughout this paper, we consider graphs with arbitrary positive edge weights representing similarities. We discuss other linkage measures in the Appendix.

A natural question is, why use similarities instead of dissimilarities? Both settings have been previously considered in the literature, and although it hard to argue that one approach is "better" than the other, our primary reason for studying similarities is because the similarity setting is arguably a more natural setting for clustering *sparse* graphs. Specifically, it is natural to assume that the similarity between pairs of vertices not connected by an edge is 0, whereas no such assignment of a dissimilarity to missing edges is obvious when edge weights represent dissimilarities.

---

[5]https://github.com/ParAlg/ParHAC

## 1.3 Related Works

ParHAC is inspired by a recent sequential $(1+\epsilon)$-approximate average-linkage HAC algorithm, which runs in $\tilde{O}(m)$ time [38]. However, obtaining a theoretically-efficient and practical parallel algorithm requires significant new ideas. The evaluation of [38] studied the difference in quality between exact and $(1 + \epsilon)$-approximate HAC and showed that even moderately small $\epsilon = 0.1$ maintains the quality of exact HAC. Our results on the quality of ParHAC are consistent with these findings.

Efficient parallelizations of HAC are known in the case of single-linkage (essentially equivalent to maximum spanning forest), and centroid linkage, if one allows $O(\log^2 n)$-approximation [57]. In other cases, the existing parallelizations of exact HAC either use linear depth [60], much larger work [28, 62] or do not come with any bounds on the running time [46, 73, 79]. The ParChain framework [79] for parallel exact HAC on pointsets recently showed that average-linkage HAC can be solved in hours for million-point datasets. Very recently, it was shown that a *distributed* implementation of the RAC algorithm can exactly cluster a billion-point dataset in a few hours using 200 machines and 3200 CPUs [73]. However, the depth of the algorithm is linear in the worst case, and can be large in practice.

In order to scale hierarchical clustering to large datasets, several HAC-inspired algorithm have been proposed, including Affinity clustering [8] and SCC [55]. Both algorithms are designed for a distributed setting, in which the number of computation rounds that one can afford is highly constrained. In particular, SCC can be seen as a best-effort approximation of HAC, given a fixed (usually small) number of rounds to run. We implemented both algorithms (using the framework that we built to develop ParHAC) and included them in our empirical evaluation.

The theoretical foundations of HAC has been developed in recent years [22, 27, 56], and have motivated using HAC in real-world settings [20, 21, 24, 25, 64]. The version of HAC that takes a graph as input has also been studied before, especially in the context of graphs derived from point sets [41, 42, 48], although without strong theoretical guarantees. Another line of work by Abboud et al. [4] showed that if the input points are in the Euclidean space and the Ward linkage method is used [78], approximate HAC can be solved in subquadratic time.

## 2 Parallel Approximate HAC

In this section we describe our parallel HAC algorithm, which we call ParHAC. We assume that the input is a weighted graph $G = (V, E, w)$, where $w : E \to \mathbb{R}^+$ gives the edge weights.

Let us now provide some background for the main ideas behind ParHAC. As observed in [73, 79], a simple exact parallel HAC algorithm can be obtained almost directly from the 40-year-old nearest-neighbor chain algorithm [9]. The parallel algorithm finds all edges $xy$ such that $xy$ is the highest-weight incident edge to each endpoint, $x$ and $y$. For simplicity, we assume here that all edge weights are distinct. One can see that these edges form a matching (which does not necessarily match all vertices), and the correctness of the nearest-neighbor chain algorithm implies that if the endpoints of these edges are merged in parallel, the output is equivalent to what a sequential HAC algorithm produces. Following [73], we call this algorithm reciprocal agglomerative clustering (RAC). Clearly, the amount of parallelism in RAC is data-dependent. In particular, it can take a linear number of steps in the worst case [73], and, as we find in our experiments, up to 21,081 steps on the YouTube (YT) real-world graph with just 1.1M vertices and 5.9M edges.

Once we consider $(1 + \epsilon)$-approximate HAC, the set of edges that we can choose to merge in the first step *grows* to include all edges whose weight is within $(1 + \epsilon)$ factor of $\mathcal{W}_{\max}$, the largest edge weight in the graph. Let us call these edges $(1+\epsilon)$-*heavy*. A major challenge is that the $(1+\epsilon)$-heavy edges no longer form a matching, and thus cannot be all merged in parallel. To see this, consider an example where all $(1 + \epsilon)$-heavy edges have a common endpoint $x$. Once one vertex merges with $x$, the size of the cluster represented by $x$ increases, which decreases the weights of all edges incident to $x$, and as a result some of these edges may cease to be $(1 + \epsilon)$-heavy.

Considering the $(1 + \epsilon)$-heavy edges motivates the use of ***geometric layering***, a technique where we group the edges into layers based on their weights and process edges in the same layer in parallel (e.g., see [10, 12]). In more detail, let $\mathcal{W}_{\max}$ and $\mathcal{W}_{\min}$ be the maximum-weight and minimum-weight in the graph, respectively. The $i$-th layer contains all edges with weight between

---

**Algorithm 1** ParHAC-ContractLayer($G = (V, E, w), T_L, \epsilon, D$)

---

**Input:** Similarity graph, $G$, threshold $T_L$, $\epsilon > 0$, dendrogram $D$.
**Ensure:** All edges in $G$ have weight $< T_L$.
 1: Let $W_{\max}$ be the current maximum-weight edge in $G$.
 2: **while** $W_{\max} \geq T_L$ **do**                                                               ▷ Outer round
 3:     Randomly color active vertices of $G$ either red or blue.
 4:     Let $R, B$ be the sets of red and blue vertices respectively.
 5:     Let $G_c = (V, E_c, w)$, where $E_c$ contains all edges in $G$ that have weight $\geq T_L$, and connect a vertex $x \in B$ with a vertex $y \in R$, where the size of $y$ is not smaller than the size of $x$.
 6:     **while** $|E_c| > 0$ **do**                                                       ▷ Inner round
 7:         Select a random priority $\pi_b$ for $b \in B$.
 8:         Let $C_b$ be a random red neighbor for each $b \in B$.
 9:         Let $T = \{(C_b, \pi_b, b) \mid b \in B\}$. Sort $T$ lexicographically.
10:         Let $T_r$ be triples from $T$ with first component equal to $r$.
11:         For each $r \in R$, select the first prefix of $T_r$, in which the total size of blue vertices exceeds $\epsilon|r|$
12:         Let $M$ be the set of (red, blue) vertex pairs selected.
13:         Merge vertices in $G$ and $G_c$ based on the pairs from $M$, updating edge weights in $G_c$.
14:         Remove edges of $G_c$ that have two red endpoints or weight below $T_L$
15:         Update $D$ based on $M$. If multiple $b \in B$ merge to a single $r \in R$, merge them into $r$ in the sorted order.
16:         Remove $r \in R$ from $G_c$ whose cluster size grew by more than a $(1 + \epsilon)$ factor since the start of the outer round.
17:     Recompute $W_{\max}$ based on the current state of $G$.

---

$((1+\epsilon)^{-(i+1)} \cdot \mathcal{W}_{\max}, (1+\epsilon)^{-i} \cdot \mathcal{W}_{\max}]$. ParHAC processes the layers one at a time; to compute the next layer it computes the maximum weight edge in the graph $W_{\max}$, and processes all edges between $((1+\epsilon)^{-1} W_{\max}, W_{\max}]$. We refer to an iteration of this loop as a ***layer-contraction phase***.

Let us now describe how to implement a layer-contraction phase. The pseudocode for the procedure is shown in Algorithm 1. The goal is to merge $(1 + \epsilon)$-heavy edges in parallel until none are left. The main challenge is in ensuring that the algorithm does not violate the approximation requirements. An ***outer-round*** of Algorithm 1 begins by randomly coloring active (i.e., non-isolated) vertices either red or blue (Line 3), assigning each color with probability $1/2$. Then, it constructs a graph $G_c$ which consists of edges of $G$ of weight belonging to the current layer (Line 5). Moreover, $G_c$ only contains edges whose endpoints have different colors, and whose red endpoint has larger size than the blue endpoint. Observe that each edge of $G$ is added to $G_c$ with probability at least $1/4$. The algorithm then performs inner-rounds while the number of edges in $G_c$ is non-zero.

Let us now describe a single ***inner-round*** (Lines 6–16). The goal of an inner round is for many blue vertices to merge into red vertices. Here we allow multiple blue vertices to merge with a single red vertex. A key property (also exploited in [38]) is that as long as the size of the cluster represented by $x$ does not grow too much within a single round, the weights of the edges incident to $x$ are very close to what they were in the beginning of the round. Specifically, assume that (in the beginning of a round) $x$ represents a cluster of size $c$. Then, until the size of this cluster exceeds $(1 + \epsilon) \cdot c$, the edges incident to $x$ that were $(1 + \epsilon)$-heavy at the beginning of the round remain (at least) $(1 + \epsilon)^2$-heavy. This allows us to merge multiple vertices with $x$ in a single round, at the cost of increasing the approximation ratio to $(1 + \epsilon)^2$ (which can be reduced to $(1 + \epsilon)$ by scaling $\epsilon$ by a constant factor).

More specifically, an inner round is implemented as follows. For each blue vertex $b \in B$, the algorithm selects a uniformly random priority $\pi_b \in [0, 1]$ (Line 7) which is used to perform symmetry breaking when merging vertices. It then chooses a random red neighbor, $C_b$, for each $b \in B$ (Line 8). If a blue vertex has no red neighbors it will not participate in the subsequent steps, but for simplicity when describing the algorithm we assume that each $b \in B$ has a valid $C_b$. The algorithm then builds $T$, a set of triples for each $b \in B$ containing the value of its candidate red neighbor $C_b$, its priority $\pi_b$, and its id, $b$ (Line 9), and sorts this set lexicographically. We call the elements of $T$ *proposals*. At this point, a red vertex may have proposals to merge from a large number of blue neighbors, and so the algorithm selects for each $r \in R$ the first prefix of $T_r$ (the merges proposing to $r$) whose total cluster size exceeds $\epsilon|r|$ (Lines 10–11). If no prefix of $T_r$ has this property, the algorithm selects all of $T_r$.

Finally, the algorithm gathers all of the $(r, b)$ vertex pairs that were selected (Line 12) and applies these merges to update both $G$ and $G_c$ (Line 13). Note that after this update we remove all edges of $G_c$ that have two red endpoints, and as a result we maintain the invariant that each edge of $G_c$ has two endpoints of distinct colors, and the size of the red endpoint is at least the size of the blue one.

## 2.1 Theoretical Analysis

We show that our algorithm performs nearly-linear total work, has poly-logarithmic depth, and thus polynomial parallelism, and has good approximation guarantees. For the purpose of this analysis we assume that $\epsilon > 0$ is a constant. We provide proofs in the Appendix and outline the main ideas here.

We start by analyzing a layer-contraction phase, and bound the total work and number of rounds required. We start by showing that within each inner round of Algorithm 1, each blue vertex makes progress in expectation either by being merged, or by having many edges incident to it be deleted.

**Lemma 2.1.** *Consider an arbitrary blue vertex $b$ within an inner round. Within this round either (a) a constant factor of edges incident to $b$ are deleted, or (b) with constant probability $b$ is merged into one of its red neighbors.*

Using this property, we can bound the number of inner rounds within an outer round to be $O(\log n)$ *whp*. Next, we show that the number of outer rounds is also $O(\log n)$ *whp*.

**Lemma 2.2.** *The number of outer rounds in a call to Algorithm 1 is $O(\log n)$ with high probability.*

The proof works by showing that for each edge $e$ of the graph $G_c$ at the beginning of the loop, either (a) the endpoints of $e$ are merged together, or (b) the weight of $e$ drops below $T_L$, or (c) an endpoint of $e$ increases its size by a factor of $(1 + \epsilon)$.

Finally, assuming the aspect-ratio $\mathcal{A} = \mathcal{W}_{\max}/\mathcal{W}_{\min} = O(\mathsf{poly}(n))$, the number of layer-contraction phases is $O(\mathsf{polylog}(n))$.[6] Putting all of the previous results together and implementing the algorithm using standard parallel primitives in the work-depth model, we obtain the following result:

**Theorem 2.3.** ParHAC *is a $(1 + \epsilon)$-approximate algorithm for average-linkage HAC that runs in $\tilde{O}(m + n)$ work in expectation and has $O(\log^4(n))$ depth with high probability.*

## 2.2 Lower Bound

We complement our upper bound with a parallel hardness result for *exact* graph-based average-linkage HAC. Since the problem is in P, we show P-hardness to show the P-completeness result. Specifically, we give an NC reduction from the P-complete monotone circuit-value problem (monotone CVP) by transforming a monotone circuit into a graph-based HAC instance such that two vertices are merged into the same cluster with a given merge similarity if and only if a target output gate evaluates to true. With minor modifications, our reduction extends to the show the P-completeness of a variant of average-linkage called WPGMA-linkage; we provide our constructions and proofs in the Appendix.

## 2.3 Algorithm Implementation

We implemented ParHAC in C++ using the *CPAM* (Compressed Parallel Augmented Maps) framework [39], which provides compressed and highly space-efficient ordered maps and sets. We build on CPAM's implementation of the Aspen framework [33, 39], which provides a compressed dynamic graph representation that supports efficient parallel updates (batch edge insertions and deletions).

**Compressed Clustered Graph Representation.** In practice ParHAC can perform a large number of rounds per-layer in the case where $\epsilon$ is small (e.g., $\epsilon = 0.01$). Although updating the entire graph on each of these rounds is theoretically-efficient (the algorithm will only perform $\tilde{O}(m + n)$ work), many of these rounds only merge a small number of vertices, and leave the majority of the edges unaffected, so updating the entire graph each round can be highly wasteful.

Instead, we designed an efficient *compressed clustered graph representation* using the CPAM framework [39] which enables us to update the underlying similarity graph in work proportional to the number of merged vertices and their incident neighbors, rather than proportional to the total

---

[6] All existing HAC approximation algorithms make similar assumptions on the aspect ratio [4, 38].

number of edges in the graph. Importantly, using CPAM enables lossless compression for integer-keyed maps to store the cluster adjacency information using just a few bytes per edge.[7] We provide more details about the representation and the supported operations in the Appendix.

**Other Hierarchical Graph Clustering Algorithms.** Our new clustered graph representation makes it very easy to implement other parallel graph clustering algorithms. In particular, we developed a faithful version of the Affinity clustering algorithm [8] and the recently proposed SCC algorithm [55] (which is essentially a thresholded version of Affinity) using a few dozens of lines of additional code. Both algorithms are essentially heuristics that are designed to mimic the behavior of HAC, while running in very few rounds (an important constraint for the distributed environments these algorithms are designed for). We note that most of the work done by these algorithms is the work required to merge clusters in the underlying graph, so by using the same primitives for merging graphs, we eliminate a significant source of differences when comparing algorithms.

# 3    Empirical Evaluation

**Experimental Setup.** We ran all of our experiments on a 72-core Dell PowerEdge R930 (with two-way hyper-threading) with $4 \times 2.4$GHz Intel 18-core E7-8867 v4 Xeon processors (with a 4800MHz bus and 45MB L3 cache) and 1TB of main memory. Our programs use a lightweight work-stealing parallel scheduler [5, 13]. Further details about our setup and input data can be found in the Appendix.

**HAC Algorithms Evaluated.** We compare ParHAC with several HAC baselines. SeqHAC is the approximate sequential average-linkage algorithm that was recently introduced [38]. ParHAC$_{\mathcal{E}}$ and SeqHAC$_{\mathcal{E}}$ are exact versions of the ParHAC and SeqHAC algorithms, where the ParHAC code takes $T_L = W_{\max}$ in each layer-contraction phase and sets $\epsilon = 0$ (i.e., the layer only consists of equal-weight edges). ParHAC$_{0.1}$ and SeqHAC$_{0.1}$ both use $\epsilon = 0.1$. We also evaluate our implementation of Affinity clustering [8] and the recently developed SCC algorithm [55], which we refer to as ParAffinity and ParSCC$_{sim}$. Both these algorithms can be thought of as heuristic parallelizations of the HAC algorithm, which are designed with speed and good parallelization properties in mind. We describe these in more detail in the Appendix. Lastly, we also compare the graph-based implementations of HAC to pointset-based HAC implementations from the scipy package using the single-, complete-, average-, and Ward-linkage measures. These are exact HAC algorithms, which look at the complete similarity matrix and thus require time which is at least quadratic in the input size.

**Building Similarity Graphs from Pointsets.** Some of our experiments generate graphs from a pointset by computing the approximate nearest neighbors (ANN) of each point, and converting the distances to similarities. We convert distances to similarities using the formula $\mathsf{sim}(u,v) = \frac{1}{1+\mathsf{dist}(u,v)}$. We then reweight the similarities by dividing each similarity by the maximum similarity. We note that we tried other similarity functions, e.g., the function $\mathsf{sim}(u,v) = \frac{1}{1+\log^k(\mathsf{dist}(u,v))}$ for small values of $k$ and obtained slightly better quality results than the original formula used in this paper. However, the relative ranking of algorithms did not change for these different functions, and as tuning the distance-to-similarity conversion seemed outside of the scope of this paper, we use the simple original formula described above in the rest of the paper. We compute the $k$-approximate nearest neighbors using a shared-memory parallel implementation of the *Vamana* approximate nearest neighbors (ANN) algorithm [45] with parameters $R = 75, L = 100, Q = \max(L, k)$. We discuss more details about the process in the Appendix.

## 3.1    Quality Evaluation

We start by investigating the quality of ParHAC with respect to ground-truth clusterings. Our goal is to understand (1) whether ParHAC preserves the clustering quality of exact average-linkage HAC (using a complete similarity matrix), and (2) what value of $\epsilon$ to use in practice. The results in this sub-section affirmatively answer (1), and show that a value of $\epsilon = 0.1$ achieves comparable quality to exact average-linkage HAC, which prior works have identified as a state-of-the-art hierarchical clustering method and use as their primary quality baseline [8, 38, 55].

---

[7]We obtain a 2.9x space savings using our CPAM-based implementation over an optimized hashtable-based implementation of a clustered graph; the running times of both implementations are essentially the same.

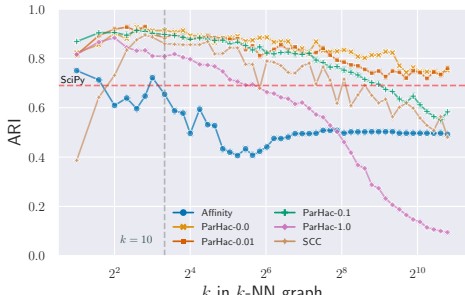
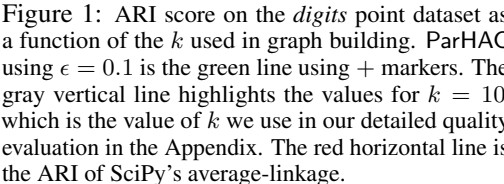
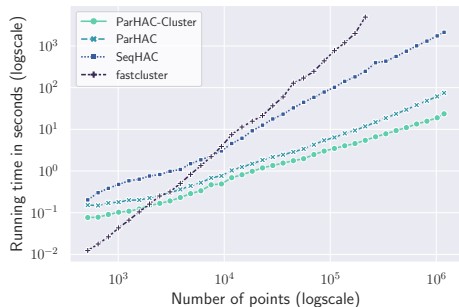

Figure 1: ARI score on the *digits* point dataset as a function of the $k$ used in graph building. ParHAC using $\epsilon = 0.1$ is the green line using $+$ markers. The gray vertical line highlights the values for $k = 10$, which is the value of $k$ we use in our detailed quality evaluation in the Appendix. The red horizontal line is the ARI of SciPy's average-linkage.

Figure 2: End-to-end running times of fastcluster's unweighted average-linkage, SeqHAC using $\epsilon = 0.1$, and ParHAC using $\epsilon = 0.1$ and 144 hyper-threads on varying-size slices of the glove-100 dataset. The time for SeqHAC and ParHAC includes the cost of solving ANN and generating the input similarity graph; ParHAC-Cluster shows only the clustering time. We terminated methods that run for more than 3 hours.

We evaluate our algorithms on the *iris*, *wine*, *digits*, and *cancer*, and *faces* classification datasets from the UCI dataset repository (found in the sklearn.datasets package). We run all of the graph-based clustering algorithms on similarity graphs generated from the input pointsets using $k = 10$ in the approximate $k$-NN construction. We run the scipy pointset clustering algorithms directly on the input pointsets. To measure quality, we use the *Adjusted Rand-Index (ARI)* and *Normalized Mutual Information (NMI)* scores, as well as the *Dendrogram Purity* measure of a hierarchical clustering [43] which we define in the Appendix. We also study the unsupervised *Dasgupta Cost* [27] measure. We give definitions of the measures in the Appendix.

**Results.** We present a table of our results in the Appendix and summarize our findings here. Our main finding is that ParHAC$_{0.1}$ achieves consistently high-quality results across all of the quality measures. For instance, for the ARI measure, ParHAC$_{0.1}$ is on average within 1.5% of the best ARI score for each graph (and achieves the best score for one of the graphs). For the NMI measure, ParHAC$_{0.1}$ is on average within 1.3% of the best NMI score for each graph (and again achieves the best score for two of the graphs). ParHAC$_{0.1}$ also achieves good results for the dendrogram purity and Dasgupta cost measures. For purity, it is on average within 1.9% of the best purity score for each graph, achieving the best score for one of the graphs, and for the unsupervised Dasgupta cost measure it is on average within 1.03% of the smallest Dasgupta cost score for each graph.

Compared with the SciPy average-linkage which is an exact HAC algorithm running on the underlying pointset, ParHAC$_{0.1}$ achieves 14.4% better ARI score on average, 3.6% better NMI score on average, 4.7% better dendrogram purity on average, and 1.02% larger Dasgupta cost on average. Compared to the best quality result obtained by either SCC$_{sim}$ or Affinity, ParHAC$_{0.1}$ consistently obtains better quality results, achieving 35.6% better ARI score on average, 12.1% better NMI score on average, 6.7% better dendrogram purity on average, and 3.1% better Dasgupta cost on average.

Overall, we find that ParHAC achieves consistently high quality results across the four quality measures that we evaluate. Our results show that being more faithful to the HAC algorithm allows ParHAC to obtain meaningful quality gains over Affinity and SCC$_{sim}$.

**Results with Varying $k$.** When converting a pointset input to the similarity setting using the $k$-NN approach as in this paper, what value of $k$ is required to achieve high quality? We studied each of the quality measures for the studied algorithms as a function of the $k$ used in the $k$-NN construction, and present our full results for each measure and each dataset in the Appendix. Here, Figure 1 shows a representative result for the ARI measure on the digits point dataset.

We find that even modest values of $k$ yield very high quality results and can *significantly outperform exact metric HAC algorithms* on the original pointset. Furthermore, even tripling, or increasing $k$ an order of magnitude either yields negligible improvement for most quality measures, or in fact degrades the quality. For example, for ParHAC with $\epsilon = 0.1$, using $k = 100$ is 10% worse than using $k = 10$, and using $k = 1000$ is 47% worse than $k = 10$. Our results suggest a twofold benefit from using a graph-based approach: (1) since small values of $k$ are sufficient for high quality results, the inputs to the clustering algorithm can be smaller, resulting in faster running times, and (2) the overall

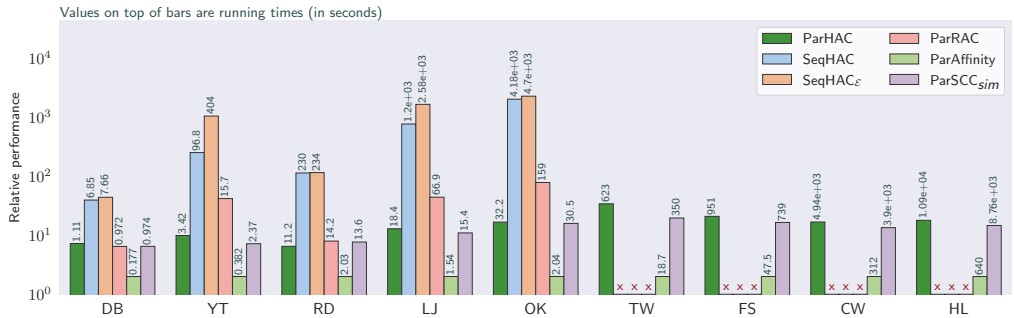

Figure 3: Relative performance of our ParHAC algorithm compared to other graph-HAC and HAC-inspired algorithms. We prefix new implementations developed in this paper with the Par prefix. The values on top of each bar show the running time of each algorithm in seconds. The times shown for the parallel algorithms are 144 hyper-thread times. We run the ParHAC and SeqHAC algorithm using $\epsilon = 0.1$. We terminated algorithms that ran longer than 6 hours and mark them with a red x.

clustering quality is higher with small inputs and can lead to significant improvements over using more similarities (or using the full dissimilarity matrix, as in pointset clustering algorithms).

## 3.2 Evaluation on Large Real-World Graphs

Next, we evaluated ParHAC's scalability on large real-world graphs, including both social network graphs, as well as sparse, high-diameter networks. We note that the Hyperlink (HL) graph is one of the largest publicly-available graphs and contains 1.7B vertices and 125B edges. Additional details and experiments can be found in the supplementary material; we summarize them:

(1) ParHAC achieves beteween 25.7–61.3x speedup over its running time on a single thread.
(2) We evaluated the running time of ParHAC when the edge weights in the graph are selected based on structural properties of the input, e.g., the degree, or the number of triangles closed by the edge. Overall, we evaluated 6 highly-different weighting schemes, and find that the gap between the fastest and slowest schemes per-graph is at most 5x.
(3) Running ParHAC with larger values of $\epsilon$ consistently results in lower running times, due to the fewer number of rounds used for large $\epsilon$.

Figure 3 shows the relative performance and of ParHAC compared with the SeqHAC and SeqHAC$_\mathcal{E}$ algorithms, and our implementations of the RAC, Affinity, and SCC$_{\text{sim}}$ algorithms. We see that our Affinity implementation is always the fastest on our graph inputs. This is due to the low theoretical and empirical round-complexity of Affinity. Compared with Affinity, SCC$_{\text{sim}}$, which runs Affinity with weight thresholding using 100 iterations is an average of 11.5x slower than Affinity due to the cost of the additional iterations. Compared to Affinity and SCC$_{\text{sim}}$, ParHAC is an average of 14.8x slower than our Affinity implementation and 1.24x slower than our SCC$_{\text{sim}}$ implementation. The speed difference between ParHAC and Affinity is due to the larger number of fine-grained rounds required by ParHAC. However, as we showed in Section 3.1, the faster running time of affinity clustering and SCC$_{\text{sim}}$ comes at the cost of significantly lower quality.

Compared with methods that implement exact or $1 + \epsilon$-approximate HAC, ParHAC is significantly faster. Compared with SeqHAC and SeqHAC$_\mathcal{E}$, ParHAC obtains 50.1x and 86.4x speedup on average respectively. Compared with RAC, ParHAC achieves an average speedup of 7.1x on the graphs that RAC can successfully complete within the time limit. Although RAC is faster than ParHAC on two of our small graph inputs, it requires a very large number of rounds on the remaining graphs.

To the best of our knowledge, our results are the first to show that graphs with tens to hundreds of billions of edges can be clustered in a matter of tens of minutes (using heuristics like Affinity and SCC$_{\text{sim}}$ with fewer iterations) to hours (using SCC$_{\text{sim}}$ with many iterations, or methods with approximation guarantees such as ParHAC). Recently, results on trillion-edge similarity graphs have been reported [55, 73]. Due to the memory requirements of storing such large datasets in memory, ParHAC may not be directly applicable; however, it may be possible to design a distributed $(1 + \epsilon)$-approximate HAC algorithm with low round-complexity by building on the ideas in this paper.

**Discussion.** To the best of our knowledge, our results are the first to show that graphs with tens to hundreds of billions of edges can be clustered in a matter of tens of minutes (using heuristic methods like Affinity and $SCC_{sim}$ with fewer iterations) to hours (using $SCC_{sim}$ with many iterations or methods with approximation guarantees such as ParHAC). We are not aware of other shared-memory clustering results that work at this large scale. Our theoretically-efficient implementations can be viewed as part of a recent line of work showing that theoretically-efficient shared-memory parallel graph algorithms can scale to the largest publicly available graphs using a modest amount of resources [31–33, 36, 66, 67].

### 3.3 Pointset Clustering: End-to-End Evaluation

We conclude by evaluating the performance of ParHAC in the case when the input is a pointset. In this setting, in order to use a graph-based HAC algorithm, one needs to construct a similarity graph before running the clustering algorithm. As a baseline, we use the average-linkage HAC implementation from fastcluster [59], which takes a pointset input. We compare with fastcluster as we found it yields more consistent (and slightly faster) performance for larger numbers of points than the implementations in sklearn and SciPy.[8] We also compare ParHAC to the SeqHAC algorithm in this setting, where SeqHAC uses the same similarity-graph building method as ParHAC described earlier in this section, and both algorithms are run using $\epsilon = 0.1$.

We run our end-to-end experiment on the Glove-100 dataset, which is a 100-dimensional dataset containing vector-embeddings for 1.18 million words. Figure 2 shows the results of our experiment. We stress that the running times of the graph-based algorithms (SeqHAC and ParHAC) include the time spent building the graph. First, we found that for $n > 3000$, the end-to-end times of ParHAC are always faster than the time taken by fastcluster, and for $n > 9400$, the end-to-end times of SeqHAC are always faster than the time taken by fastcluster. Comparing fastcluster on the largest slice of the Glove-100 dataset it can solve in under three hours with the graph-based methods, SeqHAC is 20x faster, and ParHAC is 417x faster.

## 4 Conclusion

In this paper we have introduced ParHAC, the first parallel algorithm for hierarchical agglomerative graph clustering using the average-linkage measure that has strong theoretical-bounds on its work and depth, as well as provable approximation guarantees. We have shown that ParHAC scales to massive real-world graphs with tens to hundreds of billions of edges on a single machine and achieves high-quality results compared to existing graph-based and pointset-based hierarchical clustering methods.

An interesting question for future work is whether we can reduce the round complexity of our algorithm to make it more suitable for distributed settings. One promising idea is to analyze a version of ParHAC in which we process edges from all weight buckets simultaneously, while ensuring that all merges satisfy the approximation guarantees. Another interesting question is to compare the running time and quality of bottom-up HAC methods for size-constrained clustering and balanced partitioning problems, which typically rely on top-down methods [7, 19, 30, 47]. Although it has recently been shown that it may not be possible to design algorithms for incremental and dynamic HAC with good worst-case guarantees [77], it would be interesting to design practical $(1 + \epsilon)$-approximations for these settings, potentially building on ideas from ParHAC.

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
