# A Missing Details and Proofs

We denote the degree of vertex $v$ by $deg(v)$ and use $\mathsf{Cut}(X, Y)$ to denote the set of edges between two sets of vertices $X$ and $Y$.

In this paper we focus on the *average-linkage measure* (sometimes also called *unweighted average-linkage* or *UPGMA-linkage*), which assumes that the similarity between two clusters $(X, Y)$ is equal to $\sum_{(x,y)\in\mathsf{Cut}(X,Y)} w(x,y)/(|X| \cdot |Y|)$, that is the total weight of edges between $X$ and $Y$, divided by the maximum number of possible edges between the clusters. A less common variant of average-linkage is *weighted average-linkage (WPGMA-linkage)*. For this measure, the similarity between two clusters depends not only on the current clustering, but also on the sequence of merges that created it. In particular, if a cluster $Z$ is created by merging clusters $X$ and $Y$, the similarity of the edge between $Z$ and a neighboring cluster $U$ is $\frac{\mathcal{W}(X,U)+\mathcal{W}(Y,U)}{2}$ if both edges $(X, U)$ and $(Y, U)$ exist before the merge, and otherwise just the weight of the single existing edge. We stress that *unweighted* and *weighted* in the linkage measure names refer to the linkage methods. In both cases, throughout this paper we consider graphs with arbitrary non-negative edge weights.

A linkage measure is *reducible* [9], if for any three clusters $X, Y, Z$, it holds that $\mathcal{W}(X \cup Y, Z) \leq \max(\mathcal{W}(X, Z), \mathcal{W}(Y, Z))$. We note that both average-linkage and weighted average-linkage are reducible [9].

## A.1 Sequential HAC

We start by reviewing the existing exact and $(1 + \epsilon)$-approximate sequential HAC algorithms which are two baselines that we compare against, and also the starting point for our theoretical results.

**Exact Average-Linkage.** A challenge in implementing the average-linkage HAC algorithm is to efficiently maintain edge weights in the graph as vertices (clusters) are merged. Since the average-linkage formula includes the cluster sizes of both endpoints of a $(u, v)$ edge when computing the weight of the edge, an algorithm which eagerly maintains the correct weights of all edges in the graph must update *all* of the edge weights incident to a newly merged cluster. Unfortunately, even for sparse graphs with $O(n)$ edges, one can show that such an eager approach requires $\Omega(n^2)$ time.

It was recently shown [38] that the exact average-linkage HAC algorithm can be implemented in $\tilde{O}(n\sqrt{m})$ time, using the classic nearest-neighbor chain technique [9, 58] in conjunction with a low-outdegree orientation data structure.

**Approximate Average-Linkage (**SeqHAC**).** Our parallel algorithm is inspired by a recent sequential approximate average-linkage HAC algorithm which we refer to as SeqHAC [38], which runs in near-linear time. The sequential algorithm uses a lazy strategy to avoid updating the weights of all edges incident to each merged vertex. Instead of exactly maintaining the weight of each edge, the algorithm uses the observation that the weights of edges incident to a vertex do not need to be updated every time the cluster represented by this vertex grows. Namely, if we allow an $(1 + \epsilon)^2$-approximate algorithm, the weights of edges incident to a vertex do not need to be updated until the cluster-size of this vertex grows by a multiplicative $(1 + \epsilon)$ factor. (The approximation ratio of the algorithm is squared, since the clusters represented by *both* endpoints of an edge may grow by a $(1 + \epsilon)$ factor before its weight is updated.) As this can happen at most $O(\log_{1+\epsilon} n)$ times, for a constant $\epsilon$, the overall number of edge weight updates is $\tilde{O}(m)$.

The approximate algorithm adapts the folklore *heap-based* approach [58], originally designed for an exact setting to the approximate setting. Specifically, the approximate algorithm maintains a heap keyed by the vertices, where the value assigned to a vertex is its current highest-weight incident edge (whose weight is correct up to a $(1 + \epsilon)^2$ factor). In each iteration, the highest-weight edge is chosen from the heap and its endpoints are merged. By setting parameters appropriately one can ensure that the resulting algorithm is $(1 + \epsilon)$-approximate.

## A.2 Motivating ParHAC

Recall that our approach is based on *geometric layering*, where we group the edges based on their weights and process all edges within the same layer in parallel. Let $\mathcal{W}_{\max}$ and $\mathcal{W}_{\min}$ be the maximum-weight and minimum-weight in the graph, respectively. In the geometric layering scheme,

the $i$-th layer contains all edges with weight between $((1 + \epsilon)^{-(i+1)} \cdot \mathcal{W}_{\max}, (1 + \epsilon)^{-i} \cdot \mathcal{W}_{\max}]$. The layering algorithm then processes these layers one-by-one, starting with the heaviest-weight layer. An ***active vertex*** is a vertex that has edges in the layer currently being processed by the algorithm. Under the standard assumption that the ***aspect-ratio*** of the graph is polynomially bounded, that is $\mathcal{A} = \mathcal{W}_{\max}/\mathcal{W}_{\min} = O(\mathsf{poly}(n))$, we can show that the maximum number of layers processed by the algorithm is $O(\log n)$ for any constant $\epsilon$. We note that on the real-world datasets evaluated in this paper, which include both graphs derived from real-world pointsets, and real-world graphs, the aspect ratio is at most 1000, and that all existing HAC approximation algorithms make similar assumptions on the aspect ratio [4, 38]. The key challenge for a parallel approximation algorithm based on this paradigm is that it must process all of the elements within each layer in $\mathsf{polylog}(n)$ rounds, while ensuring that it does not violate the $(1 + \epsilon)$ approximation requirements.

**Natural Approaches: Spanning Forest and Affinity.** A natural idea is to compute a spanning forest induced by the edges within the current layer, and to merge together all vertices in each tree in the forest. Using this approach, all edges within the current layer can be processed in a single spanning-forest step. Furthermore, after applying these merges all remaining edges in the graph fall into layers of smaller weight. Using a work-efficient spanning forest algorithm, the overall work and depth of this approach will be $\tilde{O}(m)$ and $O(\mathsf{polylog}(n))$, respectively [69].

A similar idea is used in the *Affinity Clustering* algorithm of Bateni et al. [8]. In this algorithm, each vertex marks its highest-weight incident edge, and the sets of vertices to be merged are the connected components of the graph formed by the marked edges. It is easy to see that this approach yields a $\tilde{O}(m)$ work and $O(\mathsf{polylog}(n))$ depth algorithm in the geometric layering scheme.

Unfortunately, both of these natural approaches fail to yield $(1 + \epsilon)$-approximate algorithms because they both choose "locally good" edges to merge without taking into account how the similarities of these edges change as they are mapped to binary merges in the dendrogram. For example, if the input graph is a path consisting of edges of the same weight, the spanning forest approach would merge all of the vertices together in one step. At the same time it is easy to see that for $\epsilon < 1/2$ an $(1 + \epsilon)$-approximate algorithm, within a single layer, must merge sets of vertices of size at most 2.

**Our Approach: Capacitated Random Mate.** Our approach is inspired by the classic random-mate technique [11] when processing each layer, but requires a careful modification to yield good approximation guarantees. Our algorithm starts by first randomly coloring the active vertices red and blue with equal probability. Directly applying the random-mate approach (e.g., as applied in the parallel connectivity algorithms of Reif [63] and Phillips [61]) would suggest merging all blue vertices into an arbitrary red neighbor. The number of edges in the layer decreases by a constant factor in expectation, thus yielding the desired work and depth bounds. Unfortunately, this approach does not yield good approximation guarantees for the same reason that the spanning forest and affinity approaches fail to do so. The issue is that the size of a red cluster can become too large, causing the similarity of a merged edge to be much smaller than $W_{\max}/(1 + \epsilon)$, where $W_{\max}$ is the *current largest edge weight*.

Our *capacitated random-mate* approach fixes this issue with the classic random-mate strategy by processing each layer in multiple *rounds* and enforcing vertex-capacities for the red vertices in each round. These capacities ensure that the sizes of clusters represented by red vertices increase by at most a $(1 + \epsilon)$ factor within a round. As a result, the weight of each edge incident to a red vertex may only decrease by a $(1 + \epsilon)$ factor.

Specifically, consider a red vertex $v$ representing a cluster $C$. Assume that at the beginning of a round all edge weights in the graph are computed exactly and the size of the cluster $C$ is $c_0$. Within the round, vertex $v$ only accepts a merge proposal from a neighboring blue vertex if the total size of the cluster $C$, including the growth incurred by previous proposals accepted within this round, is smaller than $(1 + \epsilon)c_0$. This approach ensures that the weight of any merged edge is close to $W_{\max}$, and thus lets us argue that our algorithm is $(1 + \epsilon)$-approximate. However, using this capacitated approach complicates the analysis of the round-complexity of our algorithm, making it more challenging to bound the overall work and depth. Our main theoretical contribution is to show that the capacitated approach results in a near-linear work and poly-logarithmic depth algorithm (ParHAC) that achieves a $(1 + \epsilon)$-approximation for any $\epsilon > 0$.

---

**Algorithm 2** ParHAC($G = (V, E, w), \epsilon$)

---

**Input:** Similarity graph $G$, $\epsilon > 0$.
**Ensure:** $(1 + \epsilon)$-approximate dendrogram for unweighted average-linkage HAC.
 1: Let $D$ be initialized to the identity clustering.
 2: **while** $|E| > 0$ **do**
 3:     Let $W_{\max}$ be the current maximum-weight edge in the graph.
 4:     Call ParHAC-CONTRACTLAYER$\big(G, (1 + \epsilon)^{-1} \cdot W_{\max}, \epsilon, D\big)$.
 5: **return** $D$

---

## A.3   Missing Proofs

Algorithm 2 shows the layer-based ParHAC algorithm. Next, we present the missing proofs for the work-depth and approximation analysis of the algorithm.

**Lemma A.1.** *Assuming that the aspect-ratio of the graph, $\mathcal{A} = O(\mathsf{poly}(n))$, the number of layer-contraction phases is $O(\log n)$.*

*Proof.* Since the unweighted average-linkage function is reducible, the largest weight can only decrease over the course of the algorithm. Furthermore, the smallest weight that the algorithm can encounter using the unweighted average-linkage function is proportional to $O(\mathcal{W}_{\min}/n^2)$. Thus the weight range that the algorithm runs over is $\mathcal{W}_{\max}/(\mathcal{W}_{\min}/n^2) = O(\mathsf{poly}(n))$, and only $O(\log n)$ layers are required to represent every weight in this weight range. $\square$

**Lemma 2.1.** *Consider an arbitrary blue vertex $b$ within an inner round. Within this round either (a) a constant factor of edges incident to $b$ are deleted, or (b) with constant probability $b$ is merged into one of its red neighbors.*

*Proof.* Fix the random priority $\pi_b$ for each $b \in B$. We show that the lemma follows for *any* set of distinct priorities, not necessarily random ones. It is easy to see that our algorithm is equivalent to processing the blue vertices in the order of their priorities and deciding for each of them whether it merges to a red neighbor. We say that a red vertex is *saturated* if its size has increased by more than $(1 + \epsilon)$ factor since the beginning of the outer round. Note that each saturated red vertex is removed at the end of the inner iteration. Now, consider a blue vertex $b$. If more than half of red neighbors of $b$ are saturated, we are in case (a). Otherwise, with constant probability $b$ chooses a non-saturated red neighbor and can merge with it. $\square$

**Lemma A.2.** *There are at most $O(\log n)$ inner rounds within each outer round with high probability.*

*Proof.* Fix a blue vertex $b$. There can be at most $O(\log n)$ inner rounds when the vertex satisfies case (a) of Lemma 2.1, and at most $O(\log n)$ times (with high probability) when it satisfies case (b). After that, the vertex is either merged into its red neighbor or becomes isolated. In both cases, all incident edges are removed. $\square$

**Lemma 2.2.** *The number of outer rounds in a call to Algorithm 1 is $O(\log n)$ with high probability.*

*Proof.* Consider the **while** loop in Line 6. We first prove the following claim: for each edge $e$ of the graph $G_c$ at the beginning of the loop, either (a) the endpoints of $e$ are merged together, or (b) the weight of $e$ drops below $T_L$, or (c) an endpoint of $e$ increases its size by a factor of $(1 + \epsilon)$.

To prove this fact, we observe that when the loop terminates, all edges of $G_c$ have been removed. Clearly, an edge can be removed when its endpoints are merged together or when its weight drops below $T_L$, which corresponds to cases (a) and (b). There are two more ways of removing an edge. First, an edge can be removed on Line 16 when its red endpoint grows by a factor of $(1 + \epsilon)$, which leads to case (c). Second, the edge can be removed when it connects two red vertices. This can only happen as a result of a blue vertex merging into a red vertex. However, since for each edge in $G_c$ the red endpoint has size not smaller than the blue one, whenever a blue vertex merges into a red one, the size of its cluster doubles. This finishes the proof of the claim.

From this claim, we immediately conclude that for a fixed layer, any edge $e$ of $G$ can participate in at most $O(\log n)$ inner rounds, as each endpoint of $e$ can increase its size at most $O(\log n)$ times. Since within an outer round, $e$ is included in $G_c$ with constant probability, we conclude there can be at most $O(\log n)$ outer rounds with high probability. $\qquad\square$

**Lemma A.3.** *Each outer round can be implemented in $O(m \log n + n \log^2 n)$ expected work and $O(\log^2 n)$ depth with high probability.*

*Proof.* Computing $G_c$ (Line 5) and updating $G$ and $G_c$ within (Line 13) can be done in $O(m)$ work and $O(\log n)$ depth using standard parallel primitives such as prefix-sum and filter [15]. Using a parallel comparison sort, Line 9 costs $O(n \log n)$ work and $O(\log n)$ depth [15]. The remaining steps can be implemented in $O(n)$ work and $O(\log n)$ depth using standard parallel primitives. Combining the work-depth analysis with the fact that there are $O(\log n)$ inner rounds *whp* by Lemma A.2 completes the proof. $\qquad\square$

The work-and depth component of Theorem 2.3 follows immediately from Lemma A.3, and due to the fact that there are $O(\log n)$ outer rounds (Lemma 2.2), and $O(\log n)$ layers overall (Lemma A.1). We prove the final part of Theorem 2.3 by showing that ParHAC is a good approximation to sequential average-linkage HAC.

**Theorem A.4.** ParHAC *is an $(1 + \epsilon)$-approximate average-linkage HAC algorithm.*

*Proof.* Let $1 + \delta$ be the approximation parameter used internally in the algorithm, which we will set shortly as a function of $\epsilon$. Consider the merges performed within an inner round in the algorithm. In the algorithm these merges only occur between blue vertices and their red neighbors that they are connected to with an edge with weight at least $T_L$ (the lower layer-threshold). At the start of a layer-contraction phase the maximum similarity in the graph is $W_{\max}$ and we have by construction that $(1 + \delta)^{-1} W_{\max} \leq T_L$. Furthermore, since the average-linkage function is reducible the maximum similarity in the graph throughout the rest of this layer-contraction phase is always upper-bounded by $W_{\max}$.

Consider the merges done within one inner round of the algorithm. We know that for each red vertex $r$ that participates in merges in this inner round, its cluster size never exceeds $(1 + \delta)S_I(r)$ where $S_I(r)$ is the cluster-size of $r$ at the start of the outer round. Furthermore, since we maintain the exact weights of all edges in both $G$ and $G_c$ after every inner round finishes (in Line 13), we know that the weight of each $(b_i, r)$ edge for each blue vertex $b_i$ merging into $r$ in this round is at least $T_L$. Therefore, the smallest value this weight can attain in a merge is $T_L/(1 + \delta)$, which is lower-bounded by $(1 + \delta)^{-2} W_{\max}$. Therefore, our approach yields a $(1 + \delta)^{-2}$-approximate algorithm. Now, for $\epsilon \leq 1$ we set $\delta = \epsilon/3$, and for $\epsilon > 1$, we set $\delta = \sqrt{\epsilon}/3$. Thus, we obtain an $(1 + \epsilon)$-approximate algorithm for average-linkage HAC for any $\epsilon > 0$. $\qquad\square$

# B  Parallel Hardness of Graph-Based HAC

We show hardness results for the graph-based HAC problem under two fundamental linkage measures.

## B.1  Hardness for Graph-Based HAC using Weighted Average-Linkage

We start by presenting our lower bound for weighted average-linkage measure as our lower bound for average-linkage builds on it. The problem is in P, and so we show P-hardness to show the P-completeness result. We do this by providing an NC reduction from the P-complete monotone circuit-value problem (monotone CVP).

In monotone CVP, we are given circuit consisting of gates, that has $n$ input variables $x_1, \ldots, x_n$, along with an assignment of truth values to each of the initial variables. The literals $x_i$ and $\bar{x}_i$ appear as initial truth-values set based on the truth-value of $x_i$. The circuit is monotone, so the gates consist of only AND and OR gates.

Since the topological ordering of a DAG can be computed in NC, we can assume that the circuit consists of $t$ gates $g_1, \ldots, g_t$ where each of $g_1, \ldots, g_{2n}$ corresponds to one input literal or its negation, and for $i > 2n$, gate $g_i$ has exactly two inputs, which are gates with indices less than $i$.

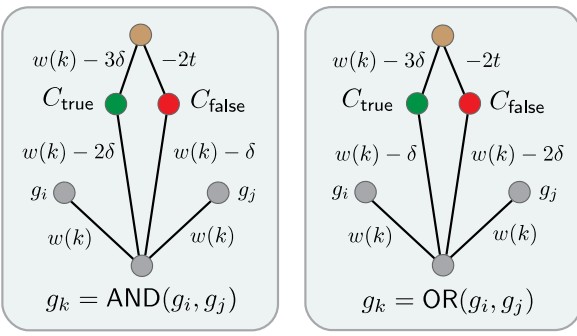

Figure 4: This figure illustrates the gadgets used in the reduction from the monotone circuit value problem to show that graph-based HAC using weighted average-linkage is P-complete.

**Construction.** The reduction builds a graph $H = (V', E')$ with $2t - 2n + 2$ vertices, where $V' = \{g_1, \ldots, g_t, C_{\text{true}}, C_{\text{false}}, d_{2n+1}, d_{2n+2}, \ldots, d_t\}$ illustrated in Figure 4. The high-level idea is to ensure that running graph-based HAC on $H$ results in gates evaluating to true merging into $C_{\text{true}}$ and gates evaluating to false merging into $C_{\text{false}}$. The only role of vertices $d_{2n+1}, d_{2n+2}, \ldots, d_t$ is to prevent $C_{\text{true}}$ and $C_{\text{false}}$ from merging with each other until the final merge. To this end, each of the vertices $d_k$ will have a positive-weight edge to $C_{\text{true}}$ and a negative-weight edge to $C_{\text{false}}$.

Note that the construction can be easily modified not to use negative weights, since in the case of weighted average-linkage increasing all edge weights by the same amount does not affect the course of the algorithm.

The construction is entirely local and can thus be easily parallelized using standard parallel primitives such as prefix-sums. The graph we define adds a gadget for each gate, which is defined using $w(i) := t - i$, and a constant $\delta \in (0, 1/4)$.

The graph $H$ is constructed by adding the following set of edges. For each $k = 1, \ldots, t$:

1. If $k \leq 2n$, we add an edge of weight $w(0)$ between $g_k$ and $C_{\text{true}}$ or $C_{\text{false}}$, depending of the value of the literal $g_k$.

2. Assume $k > 2n$ and $g_k = \text{AND}(g_i, g_j)$. We add the following edges:
   $\{(g_k, C_{\text{true}}, w(k) - 2\delta), (g_k, C_{\text{false}}, w(k) - \delta), (g_k, g_i, w(k)), (g_k, g_j, w(k)), (d_k, C_{\text{true}}, w(k) - 3\delta), (d_k, C_{\text{false}}, -2t)\}$.

3. Assume $k > 2n$ and $g_k = \text{AND}(g_i, g_j)$. We add the following edges:
   $\{(g_k, C_{\text{true}}, w(k) - \delta), (g_k, C_{\text{false}}, w(k) - 2\delta), (g_k, g_i, w(k)), (g_k, g_j, w(k)), (d_k, C_{\text{true}}, w(k) - 3\delta), (d_k, C_{\text{false}}, -2t)\}$.

Finally, we add an edge between $C_{\text{true}}$ and $C_{\text{false}}$ of weight $-t$. An illustration of the resulting gadgets built from $g_k$ for AND and OR gates is shown in Figure 4. We first show that the gadgets are processed in the order of their indices.

**Lemma B.1.** *Consider running the graph-based HAC algorithm with weighted average-linkage on $H$. For any $k \geq 2n$, the algorithm begins by merging gates $g_1, \ldots, g_k$ and $d_{2n+1}, \ldots, d_k$ into $C_{\text{true}}$ or $C_{\text{false}}$, before merging any gates with indices larger than $k$. Moreover, once all gates of indices up to $k$ have been merged, the weight of the edge between $C_{\text{true}}$ and $C_{\text{false}}$ is at most $-t$.*

*Proof.* We prove the claim by induction on $k$. To show the base case, we need to show that the algorithm begins by merging the gates corresponding to literals to either $C_{\text{true}}$ or $C_{\text{false}}$.

Observe that $w(0)$, the edge weight used for connecting literals to $C_{\text{true}}$ or $C_{\text{false}}$, is the largest edge weight in the graph. Moreover, each merge of a literal gate reduces by 1 the number of edges of weight at least $w(0)$, so the algorithm indeed begins by merging gates $g_1, \ldots, g_{2n}$ into $C_{\text{true}}$ or $C_{\text{false}}$.

Consider now $k > 2n$ and assume the algorithm has merged $g_1, \ldots, g_{k-1}$ and $d_{2n+1}, \ldots, d_{k-1}$ into $C_{\text{true}}$ and $C_{\text{false}}$. Hence, the remaining set of vertices are $g_k, \ldots, g_t, d_k, \ldots, d_t, C_{\text{true}}$, and $C_{\text{false}}$.

Let us look at the set of edge weights in the remaining graph.

- For any $d_j$, where $j \geq k$, clearly the incident edges still have weights $w(j) - 3\delta$ and $-4t$.

- For any $g_j$, where $j \geq k$, by induction hypothesis, $g_j$ has not yet participated in any merges. In the original graph the edges incident to $g_j$ connect $g_j$ to

    1. its inputs (weight $w(j)$),
    2. $C_{\text{true}}$ and $C_{\text{false}}$ (weight at least $w(j) - 2\delta$),
    3. gates with indices larger than $j$ (of weight at most $w(j) - 1$).

    By the induction hypothesis, the only change that may have happened to the set of edges incident to $g_j$ is that edges of type 1 above *may* have been merged with edges of type 2. Still, $g_j$ has an incident edge of weight belonging to $(w(j) - 2\delta, w(j)]$.

- In addition, there is an edge between $C_{\text{true}}$ and $C_{\text{false}}$ of negative weight.

From the definition of $w(\cdot)$ it follows that the only edges of weight more than $w(k) - 1$ are incident to $g_k$ and $d_k$. Moreover, the highest weight edge is incident to $g_k$. Since both inputs to $g_k$ have been already merged into $C_{\text{true}}$ or $C_{\text{false}}$, it follows that $g_k$ merges into one of them as well. After that, $d_k$ merges into $C_{\text{true}}$.

Before these two steps, the edge between $C_{\text{true}}$ and $C_{\text{false}}$ had weight of at most $-t$. Since $(w(k) - t)/2 \leq 0$, after the first merge, the weight between $C_{\text{true}}$ and $C_{\text{false}}$ remains negative. Denote it by $x$. After $d_k$ is merged into $C_{\text{true}}$, the weight of the edge between $C_{\text{true}}$ and $C_{\text{false}}$ is $(x - 2t)/2 \leq -2t/2 = -t$. □

**Lemma B.2.** *Consider running graph-based HAC algorithm with weighted average-linkage on $H$ until no edges of nonnegative weight remain. Then, the algorithm produces exactly two clusters, where one of them contains $C_{\text{true}}$ and vertices corresponding to gates that evaluate to* true *and the other one contains $C_{\text{false}}$ and all vertices corresponding to gates that evaluate to* false.

*Proof.* We show that gates $g_1, \ldots, g_k$ are correctly merged into $C_{\text{true}}$ or $C_{\text{false}}$ using an induction on $k$. For $k \leq 2n$ this follows directly from the construction and Lemma B.1.

Now assume $k > 2n$. We use Lemma B.1 and look at the point where $g_k$ is merged for the first time. At this point, we know that the inputs to $g_k$ have been (correctly) merged into $C_{\text{true}}$ or $C_{\text{false}}$.

The proof is a case-by-case analysis, depending on the gate type and the values of the inputs. For example, if $g_k = \text{AND}(g_i, g_j)$, and $g_i = g_j = \text{true}$, we have that the updated weight of $(g_k, C_{\text{true}}) = 1 - \delta/2$, and the weight of $(g_k, C_{\text{false}}) = 1 - \delta$, and therefore $g_k$ is correctly merged with $C_{\text{true}}$. On the other hand, if one or both of $\{g_i, g_j\} = \text{false}$, then the updated weight of $(g_k, C_{\text{true}}) \leq 1 - \delta$, and the weight of $(g_k, C_{\text{false}}) \geq 1 - \delta/2$, ensuring that $g_k$ is correctly merged with $C_{\text{false}}$. It is easy to check the other cases similarly.

It follows from Lemma B.1 that the clusters containing $C_{\text{true}}$ and $C_{\text{false}}$ are never merged with each other. The Lemma follows. □

This immediately implies the following.

**Theorem B.3.** *Graph-based HAC using weighted average-linkage is* P-*complete.*

### B.2   Hardness for Graph-Based HAC using Average-Linkage

Our reduction for average-linkage is similar to the reduction for weighted average-linkage, but requires additional steps to handle how average-linkage adjusts the weight of an edge between two clusters based on their respective sizes. [9]

As before, our reduction starts from the monotone CVP problem, and the idea is to have HAC simulate the execution of the circuit so that gates evaluating to true (false) are merged to $C_{\text{true}}$ ($C_{\text{false}}$).

One of the main problems to deal with is that the weight of an edge between two clusters $U$ and $V$ in average-linkage is equal to the sum of the weights crossing the cut from $U$ to $V$ normalized by the

---

[9]Note that average-linkage is *not* a special case of weighted average-linkage measure. See the start of Appendix A for the definitions.

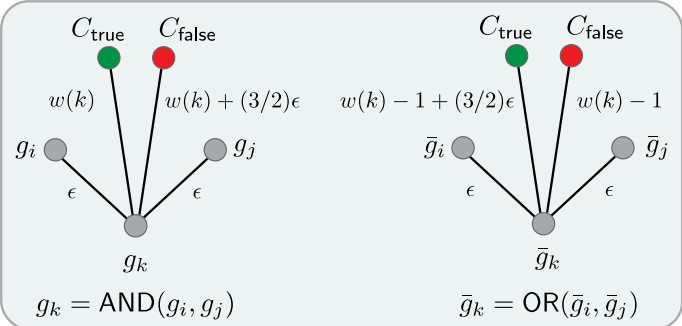

Figure 5: Gadgets used in the reduction from the monotone circuit value problem to show that graph-based HAC using average-linkage is P-complete.

product $|U| \cdot |V|$. The challenge is that we do not know (without running the algorithm) what the size of $C_{\text{true}}$ or $C_{\text{false}}$ will be at some intermediate step in the algorithm, since these sizes depend on the number of gates that evaluate to true and false respectively.

To fix this problem, we add *two vertices per gate* corresponding to $g_i$ and $\bar{g}_i$ and ensure that if $g_i$ merges to $C_{\text{true}}$, $\bar{g}_i$ merges to $C_{\text{false}}$. In addition, we need to assure that $C_{\text{true}}$ and $C_{\text{false}}$ do not merge with each other. To this end, we ensure that the graph-based HAC algorithm starts by increasing the clusters of both $C_{\text{true}}$ and $C_{\text{false}}$ to a large size, which makes them harder to merge with each other.

**Construction.** As in the case of weighted average linkage, we can assume that the circuit consists of $t$ gates $g_1, \ldots, g_t$ where each of $g_1, \ldots, g_{2n}$ corresponds to one input literal or its negation, and for $i > 2n$, gate $g_i$ has exactly two inputs, which are gates with indices less than $i$. The reduction builds a graph $H = (V', E')$, which initially has two special vertices, $C_{\text{true}}$ and $C_{\text{false}}$. In the description we use $w(k) := 4t - 2k$ and $\epsilon = 1/100$.

First, we add $(1/\epsilon - 3) \cdot t$ dummy vertices, and connect half of them to $C_{\text{true}}$ and half to $C_{\text{false}}$ using edges of weight $t^3$.

Then, we add $2n$ vertices corresponding to the literals, and connect each of them to either $C_{\text{true}}$ or $C_{\text{false}}$, depending on the literal value, also with an edge of weight $t^3$. These high weight edges ensure that the graph-based HAC algorithm proceeds by merging the dummy vertices and literal vertices to $C_{\text{true}}$ or $C_{\text{false}}$.

Finally, we add $2(t - 2n)$ vertices corresponding to the gates. For a fixed $i$ ($2n < i \le t$), we add two vertices $g_i$ and $\bar{g}_i$ corresponding to gate $g_i$ in the circuit.

1. For $g_k = \text{AND}(g_i, g_j)$, create a vertex $g_k$ with edges:
   $\{(g_k, C_{\text{true}}, w(k)), (g_k, C_{\text{false}}, w(k) + (3/2)\epsilon)\}$ and
   $\{(g_k, g_i, \epsilon), (g_k, g_j, \epsilon))\}$.

2. For $\bar{g}_k = \text{OR}(\bar{g}_i, \bar{g}_j)$, create a vertex $\bar{g}_k$ with edges:
   $\{(\bar{g}_k, C_{\text{true}}, w(k) - 1 + (3/2)\epsilon), (\bar{g}_k, C_{\text{false}}, w(k) - 1)\}$ and
   $\{(\bar{g}_k, \bar{g}_i, \epsilon), (\bar{g}_k, \bar{g}_j, \epsilon)\}$.

The construction for $g_k = \text{AND}(g_i, g_j)$ swaps $g_k$ and $\bar{g}_k$ in the descriptions above. Induction over the merge steps, performing case analysis on the truth values of the inputs to each gate, similar to our proof for weighted average-linkage yields the following theorem.

**Theorem B.4.** *Graph-based HAC using average-linkage is* P-*complete.*

*Proof.* Let a merge step of the graph-based HAC algorithm be two consecutive merges that it performs, and call the internal merges sub-merges. The proof shows by induction on the merge steps that the next merge merges the vertices $g$ and $\bar{g}$, and that the merge correctly simulates the circuit logic for $g$ and $\bar{g}$.

Suppose in the $k$-th step that some vertex $g_{k'}$, with $k' > k$ is merged instead of $g_k$ (the proof is essentially identical for $\bar{g}_k$). We consider two cases. In the first case, suppose the merge is between $g_{k'}$ and either $C_{\text{true}}$ or $C_{\text{false}}$. Denote the weights of $C_{\text{true}}$ and $C_{\text{false}}$ by $r$. The weight of the edge

between $g_k$ and $C_{\text{true}}$ or $C_{\text{false}}$ is at least $w(k)/r$ and the weight of any edge incident to $g_{k'}$ is at most $(w(k') + (5/2)\epsilon)/r < w(k)/r$, a contradiction.

In the second case, suppose the merge is between $g_{k'}$ and one of its inputs, $g_j$, which clearly uses an edge of weight at most $\epsilon$. To argue that this cannot happen, we show that the linkage similarity of $g_k$ to both $C_{\text{true}}$ and $C_{\text{false}}$ is greater than $\epsilon$. Indeed, the cluster size of both $C_{\text{true}}$ and $C_{\text{false}}$ is at most $1 + t(1/\epsilon - 3) + 2t < t/\epsilon$. Hence, the weight of an edge from $g_k$ to any of these clusters is at least $(w(k) - 1)/(t/\epsilon) \leq (w(t) - 1)/(t/\epsilon) = (4t - 2t - 1)/(t/\epsilon) > \epsilon$.

Finally, we also show that $C_{\text{true}}$ and $C_{\text{false}}$ do not merge with each other. Observe that the sizes of both these clusters are at least $t/(2\epsilon)$. The total weight between them can be upper bounded by the total weight of edges connecting $C_{\text{true}}$ and $C_{\text{false}}$ to the vertices representing gates, which is at most $4 \cdot t \cdot w(0) \leq 16t^2$. Hence, the linkage similarity is at most $16t^2/(t^2/\epsilon^2) = 16\epsilon^2 < \epsilon$.

Next, we show that the gadgets correctly simulate the gates' logic. Suppose $g_k = \mathsf{AND}(g_i, g_j)$ and $\bar{g}_k = \mathsf{OR}(\bar{g}_i, \bar{g}_j)$. Since the edges incident to $g_k$ have higher weights than edges incident to $\bar{g}_k$, $g_k$ are merged first. Below, we consider the *unnormalized* weights, i.e., the weights before normalizing by the product of the cluster weights (comparing unnormalized weights is sufficient since clusters of $C_{\text{true}}$ and $C_{\text{false}}$ have the same size). Suppose $g_i = g_j = \mathsf{false}$, then the weight between $g_k$ and $C_{\text{true}}$ is $w(k)$, the weight to $C_{\text{false}}$ is $w(k) + (3/2)\epsilon + 2\epsilon$ and $g_k$ is merged to $C_{\text{false}}$. If only one of $g_i = \mathsf{false}$, then the weight between $g_k$ and $C_{\text{true}}$ is $w(k) + \epsilon$, the weight to $C_{\text{false}}$ is $w(k) + (3/2)\epsilon + \epsilon$ and $g_k$ will also be merged to $C_{\text{false}}$. Finally, if $g_i = g_j = \mathsf{true}$, the weight between $g_k$ and $C_{\text{true}}$ is $w(k) + 2\epsilon$, the weight to $C_{\text{false}}$ is $w(k) + (3/2)\epsilon$ and $g_k$ is merged to $C_{\text{true}}$.

Note that when performing the merge for $\bar{g}_k$, the cluster that $\bar{g}_k$ must not merge to has size one larger than the cluster it should merge to. It is easy to calculate as in the example above that this imbalance in sizes does not affect the correctness of the gadget for $\bar{g}_k$.

$\square$

## C  ParHAC **Implementation**

This section completes our implementation description introduced in Section C. We implemented ParHAC in C++ using the recently developed *CPAM* (Compressed Parallel Augmented Maps) framework [39], which lets users construct compressed and highly space-efficient ordered maps and sets. We build on CPAM's implementation of the Aspen framework [33, 39], which provides a lossless compressed dynamic graph representation that supports efficient updates (batch edge insertions and deletions). We also use the ParlayLib library for parallel primitives such as reduction, prefix-sum, and sorting [13].

**Geometric Layering.** Instead of explicitly extracting the edges in each layer and constructing the graph $G_c$ in each outer round, we opted for a simpler approach which works as follows: for each layer-contraction phase in the algorithm we first (i) compute the set of vertices that are active in this phase, i.e., have at least one incident edge in the current layer and then (ii) repeatedly run the capacitated random-mate steps for an inner round from Algorithm 1, recomputing the weights of *all affected edges* in the graph at the end of each inner round. Our implementation modifies Algorithm 1 by removing Line 6, which effectively fuses the outer and inner rounds. After each round, we check whether each vertex is still active, and continue within this layer until no further active vertices remain.

**Compressed Clustered Graph Representation.** We observed that in practice our ParHAC algorithm can perform a large number of rounds per-layer in the case where $\epsilon$ is very small (e.g., $\epsilon = 0.01$). Although updating the entire graph on each of these rounds is theoretically-efficient (the algorithm will only perform $\tilde{O}(m + n)$ work), many of these rounds only merge a small number of vertices, and leave the majority of the edges unaffected. Hence, updating the weights of all edges in each round can be highly wasteful.

Instead, we designed an efficient *compressed clustered graph representation* using purely-functional compressed trees from the CPAM framework [39]. The new data structure handles all merge operations that affect the underlying similarity graph in work proportional to the number of merged vertices and their incident neighbors rather than proportional to the total number of edges in the graph.

Importantly, using CPAM enables lossless compression for integer-keyed maps to store the cluster adjacency information using just a few bytes per edge.[10]

Our data structure stores the vertices in an array, and stores the current neighbors of each vertex in a weight-balanced compressed purely-functional tree called a PaC-tree [39]. Each vertex also stores several extra variables that store its current cluster size, and variables that help build the dendrogram, such as the current cluster-id of each vertex. There are two key operations that we provide in the compressed clustered graph representation:

(1) *MultiMerge*: given a sequence $M$ of $(r, b)$ vertex merges where $r$ is a red vertex and $b$ is a blue vertex, update the graph based on all merges in $M$.
(2) *Neighborhood Primitives*: apply a function $f$ (e.g., map, reduce, filter) in parallel over all current neighbors of a vertex.

For example, our implementation of Algorithm 1 uses a parallel map operation over the neighbors of all blue vertices to generate a sequence of merges, which it then supplies to the MultiMerge primitive. Additionally, in our implementation, all of the details of maintaining the dendrogram, keeping track of the size of each cluster, and maintaining the current state of the underlying weighted similarity graph are handled by the compressed clustered graph representation. This enables us to write high-level code for our ParHAC implementation while handling the more low-level details about efficiently merging vertices within the clustered graph code.

**Other Parallel Graph Clustering Algorithms.** Our new graph representation makes it very easy to implement other parallel graph clustering algorithms. In particular, we developed a faithful version of the Affinity clustering algorithm [8] and the recently proposed SCC algorithm [55] (which is essentially a thresholded version of Affinity) using a few dozens of lines of additional code. Both algorithms are essentially heuristics that are designed to mimic the behavior of HAC, while running in very few rounds (an important constraint for the distributed environments these algorithms are designed for). We note that most of the work done by these algorithms is the work required to merge clusters in the underlying graph, and so by using the same primitives for merging graphs, we eliminate a significant source of differences when comparing algorithms.

The Affinity clustering algorithm [8] is inspired by Borůvka's MST algorithm [18]. In each round of Affinity clustering, each vertex selects its heaviest incident edge, and all connected components induced by the chosen edges are merged to form new clusters. This process continues until no further edges remain in the graph. After computing best edges, our implementation uses a highly optimized parallel union-find connectivity algorithm from the ConnectIt framework [34, 44] to compute a unique vertex per Affinity tree, which is used as a 'red' vertex in the MultiMerge procedure, with all other vertices in its component being 'blue' vertices.

The SCC algorithm [55] is closely related to the Affinity algorithm, and can be viewed as running Affinity with different weight thresholds. Specifically, in the $i$-th round, the SCC algorithm runs a round of Affinity on the graph induced by all edges with weight at least $T_i$ where $T_i$ is the weight threshold on the $i$-th round. Given the maximum number of rounds $R$, and the upper threshold $U$ and lower threshold $L$, the SCC algorithm runs a sequence of $R$ rounds where $T_i = U \cdot (L/U)^{R-i}$. As with Affinity, if the graph becomes empty before all $R$ iterations are run, the algorithm terminates early. We note that our implementation is a best-effort approximation of the SCC algorithm, since SCC was originally designed for the dissimilarity setting and there is no one-to-one mapping to the similarity setting. We refer to our implementation of SCC as $\mathrm{SCC_{sim}}$.

We note that our initial implementations of Affinity and $\mathrm{SCC_{sim}}$ were developed in the GBBS framework for static graph processing [32, 37], and did not make use of the compressed clustered graph representation. These initial implementations recomputed the weights of *all* edges in the graph in each round. We found that our new implementations using the compressed clustered graph, which only recompute the weights of edges incident to a merge, are between 7–11x faster across the graphs we evaluate.

---

[10]We find a 2.9x improvement in space-usage by using our CPAM-based implementation over an optimized hashtable-based implementation of a clustered graph, while the running times of both implementations are essentially the same.

# D   Experimental Results

**Graph Data.** We list information about graphs used in our experiments in Table 1. ***com-DBLP (DB)*** is a co-authorship network sourced from the DBLP computer science bibliography (License: *CC BY-SA*). ***YouTube (YT)*** is a social-network formed by user-defined groups on the YouTube site (License: *CC BY-SA*). ***LiveJournal (LJ)*** is a directed graph of the social network (License: *CC BY-SA*). ***com-Orkut (OK)*** is an undirected graph of the Orkut social network (License: *CC BY-SA*). ***Friendster (FS)*** is an undirected graph describing friendships from a gaming network (License: *CC0 1.0*). All of the aforementioned graphs are sourced from the SNAP dataset [52].[11] The licenses are obtained using the data licensing information at the Network Repository.[12] ***USA-Road (RD)*** is an undirected road network from the DIMACS challenge [29] (License: *CC BY-SA*).[13] ***Twitter (TW)*** is a directed graph of the Twitter network, where edges represent the follower relationship [50] (License: *CC BY-SA*). [14] ***ClueWeb (CW)*** is a web graph from the Lemur project at CMU [17] (the authors use a custom open-source license that "provide flexibility to scientists and software developers"; for more details please see `http://www.lemurproject.org/`). [15] ***Hyperlink (HL)*** is a hyperlink graph obtained from the WebDataCommons dataset where nodes represent web pages [54] (License: available to anyone following the Common Crawl Terms of Use: `https://commoncrawl.org/terms-of-use/`). [16] We note that the large real-world graphs that we study are not weighted, and so we set the similarity of an edge $(u, v)$ to $\frac{1}{\log(deg(u)+deg(v))}$. For the CW and HL graphs, which contain tens to hundreds of billions of edges, due to memory constraints on the machine we use, we set the *initial edge weights* to 1 and use byte-codes to store the weights in a number of bytes proportional to their size [39, 70]. We emphasize that the *aggregate weights on edges* (the number of edges in the original graph that each edge between clusters represents) as the algorithm progresses grow *significantly larger*, and do not simply stay fixed at 1.

We also consider graphs generated from a pointset by computing the approximate nearest neighbors (ANN) of each point, and converting the distances to similarities. Our graph building process converts distances to similarities using the formula $\text{sim}(u, v) = \frac{1}{1+\text{dist}(u,v)}$.[17] It then reweights similarities by dividing each similarity by the maximum similarity. In our implementation, we compute the $k$-approximate nearest neighbors using a shared-memory parallel implementation of the *Vamana* approximate nearest neighbors (ANN) algorithm [45] with parameters $R = 75, L = 100, Q = \max(L, k)$. We note that this parameter setting yields almost perfect recall on the SIFT datasets for the 10-nearest neighbors. More details about the quality of the Vamana algorithm can be found on ANN-Benchmarks [6]. The datasets that we use are sourced from the sklearn.datasets package[18] These datasets are originally sourced from the UCI repository [19], which does not have a clearly stated licensing policy. However, we note that all of the datasets we test on have been widely-used in the machine learning literature, with datsets such as Iris being used in thousands of publications to date. and the Glove-100 dataset from the ANN-Benchmarks collection [6]. The Glove-100 dataset is licensed as Public Domain Dedication and License v1.0.[20] We symmetrized all directed graph inputs studied in this paper.

**Experimental Setup.** We ran all of our experiments on a 72-core Dell PowerEdge R930 (with two-way hyper-threading) with $4 \times 2.4$GHz Intel 18-core E7-8867 v4 Xeon processors (with a 4800MHz bus and 45MB L3 cache) and 1TB of main memory. Our programs use a lightweight parallel scheduler based on the Arora-Blumofe-Plaxton deque [5, 13]. For parallel experiments, we use `numactl -i all` to balance the memory allocations across the sockets.

---

[11]Sources: `https://snap.stanford.edu/data/`.

[12]https://networkrepository.com/policy.php

[13]http://www.dis.uniroma1.it/challenge9/

[14]Source: `http://law.di.unimi.it/webdata/twitter-2010/`.

[15]Source: `https://law.di.unimi.it/webdata/clueweb12/`.

[16]Source: `http://webdatacommons.org/hyperlinkgraph/`.

[17]We also considered other distance-to-similarity schemes, e.g., $\text{sim}(u, v) = 1/(1 + \log^c(\text{dist}(u, v)))$ and $\text{sim}(u, v) = e^{-\text{dist}(u,v)}$. We observed that the first choice, with $c > 1$ yielded slightly better quality results for sparse graph-based methods, but chose the scheme used in this paper for its simplicity.

[18]For more detailed information see `https://scikit-learn.org/stable/datasets.html`.

[19]`https://archive.ics.uci.edu/ml/datasets.php`

[20]`https://nlp.stanford.edu/projects/glove/`

Table 1: Graph inputs, including the number of vertices $(n)$, number of directed edges $(m)$, and the average degree $(m/n)$.

| Graph Dataset | Num. Vertices | Num. Edges | Avg. Degree |
|---|---|---|---|
| *com-DBLP* (**DB**) | 425,957 | 2,099,732 | 4.92 |
| *YouTube-Sym* (**YT**) | 1,138,499 | 5,980,886 | 5.25 |
| *USA-Road* (**RD**) | 23,947,348 | 57,708,624 | 2.40 |
| *LiveJournal* (**LJ**) | 4,847,571 | 85,702,474 | 17.6 |
| *com-Orkut* (**OK**) | 3,072,627 | 234,370,166 | 76.2 |
| *Twitter* (**TW**) | 41,652,231 | 2,405,026,092 | 57.7 |
| *Friendster* (**FS**) | 65,608,366 | 3,612,134,270 | 55.0 |
| *ClueWeb* (**CW**) | 978,408,098 | 74,744,358,622 | 76.3 |
| *Hyperlink* (**HL**) | 1,724,573,718 | 124,141,874,032 | 71.9 |

### D.1 Quality Evaluation

**Quality Metrics.** To measure quality, we use the ***Adjusted Rand-Index (ARI)*** and ***Normalized Mutual Information (NMI)*** scores, which are standard measures of the quality of a clustering with respect to a ground-truth clustering. We also use the ***Dendrogram Purity*** measure [43], which takes on values between $[0, 1]$ and takes on a value of $1$ if and only if the tree contains the ground truth clustering as a tree consistent partition (i.e., each class appears as exactly the leaves of some subtree of the tree). Given a tree $T$ tree with leaves $V$, and a ground truth partition of $V$ into $C = \{C_1, \ldots, C_l\}$ classes, define the purity of a subset $S \subseteq V$ with respect to a class $C_i$ to be $\mathcal{P}(S, C_i) = |S \cap C_i|/|S|$. Then, the purity of $T$ is

$$\mathcal{P}(T) = \frac{1}{|Pairs|} \sum_{i=1}^{l} \sum_{x,y \in C_i, x \neq y} P(\mathsf{lca}_T(x, y), C_i)$$

where $Pairs = \{(x, y) \mid \exists i \text{ s.t. } \{x, y\} \subseteq C_i\}$ and $\mathsf{lca}_T(x, y)$ is the set of leaves of the least common ancestor of $x$ and $y$ in $T$. Lastly, we also study the unsupervised ***Dasgupta Cost*** [27] measure of our dendrograms, which is measured with respect to an underlying similarity graph $G(V, E, w)$ and is defined as:

$$\sum_{(u,v) \in E} |\mathsf{lca}_T(u, v)| \cdot w(u, v)$$

**Quality Study.** Table 2 shows the results of our quality study. We observe that our ParHAC$_{0.1}$ algorithm achieves consistent high-quality results across all of the quality measures that we evaluate. For the ARI measure, ParHAC$_{0.1}$ is on average within 1.5% of the best ARI score for each graph (and achieves the best score for one of the graphs). For the NMI measure, ParHAC$_{0.1}$ is on average within 1.3% of the best NMI score for each graph (and again achieves the best score for two of the graphs). ParHAC$_{0.1}$ also achieves good results for the dendrogram purity and Dasgupta cost measures. For purity, it is on average within 1.9% of the best purity score for each graph, achieving the best score for one of the graphs, and for the unsupervised Dasgupta cost measure it is on average within 1.03% of the smallest Dasgupta cost score for each graph. Compared with the SciPy unweighted average-linkage which runs on the underlying pointset, ParHAC$_{0.1}$ achieves 14.4% better ARI score on average, 3.6% better NMI score on average, 4.7% better dendrogram purity on average, and 1.02% larger Dasgupta cost on average. Lastly, we observe that out of Affinity and SCC$_{\text{sim}}$, SCC$_{\text{sim}}$ nearly always outperforms Affinity on all quality measures. Compared to SCC$_{\text{sim}}$, ParHAC$_{0.1}$ consistently obtains better quality results, achieving 35.6% better ARI score on average, 12.1% better NMI score on average, 6.7% better dendrogram purity on average, and 3.1% better Dasgupta cost on average.

**Quality vs. $k$.** We also show additional results for the quality of different algorithms studied in this paper versus the value of $k$ used in the $k$-NN graph construction. For clarity, the following figures are shown at the end of the appendix.

- Figures 15–18 show quality measures for the *iris* dataset.
- Figures 19–22 show quality measures for the *wine* dataset.
- Figures 23–26 show quality measures for the *digits* dataset.

Table 2: Adjusted Rand-Index (ARI), Normalized Mutual Information (NMI), Dendrogram Purity, and Dasgupta cost of our new ParHAC implementations (columns 2–3) versus the RAC and SeqHAC implementations (columns 4–5), our Affinity and $SCC_{sim}$ implementations (columns 6–7), and the HAC implementations from SciPy (columns 8–11). Both RAC and $SeqHAC_\varepsilon$ are exact HAC algorithms, and thus compute the same dendrogram. The scores are calculated by evaluating the clustering generated by each cut of the dendrogram against ground-truth labels. All graph-based implementations are run over the similarity graph given by an approximate $k$-NN graph with $k = 10$. The Dasgupta cost is computed over the complete similarity graph generated from the all-pairs distance graph. The best quality score for each graph is in green and underlined.

| | Dataset | $ParHAC_\varepsilon$ | $ParHAC_{0.1}$ | RAC and $SeqHAC_\varepsilon$ | $SeqHAC_{0.1}$ | Affinity | $SCC_{sim}$ | Sci-Single | Sci-Complete | Sci-Avg | Sci-Ward |
|---|---|---|---|---|---|---|---|---|---|---|---|
| ARI | iris | 0.892 | 0.911 | 0.873 | 0.873 | 0.599 | 0.786 | 0.715 | 0.642 | 0.759 | 0.731 |
| | wine | 0.401 | 0.401 | 0.401 | 0.401 | 0.416 | 0.374 | 0.298 | 0.371 | 0.352 | 0.368 |
| | digits | 0.912 | 0.896 | 0.895 | 0.895 | 0.625 | 0.851 | 0.661 | 0.479 | 0.690 | 0.813 |
| | cancer | 0.440 | 0.491 | 0.447 | 0.447 | 0.375 | 0.197 | 0.561 | 0.465 | 0.537 | 0.406 |
| | faces | 0.621 | 0.618 | 0.610 | 0.610 | 0.460 | 0.607 | 0.468 | 0.472 | 0.529 | 0.608 |
| NMI | iris | 0.858 | 0.876 | 0.842 | 0.842 | 0.692 | 0.780 | 0.734 | 0.722 | 0.806 | 0.770 |
| | wine | 0.409 | 0.409 | 0.394 | 0.394 | 0.426 | 0.400 | 0.410 | 0.442 | 0.428 | 0.428 |
| | digits | 0.928 | 0.916 | 0.913 | 0.931 | 0.768 | 0.859 | 0.562 | 0.711 | 0.830 | 0.869 |
| | cancer | 0.445 | 0.464 | 0.445 | 0.445 | 0.412 | 0.325 | 0.316 | 0.428 | 0.456 | 0.423 |
| | faces | 0.879 | 0.874 | 0.873 | 0.873 | 0.858 | 0.867 | 0.848 | 0.849 | 0.861 | 0.869 |
| Purity | iris | 0.931 | 0.943 | 0.920 | 0.920 | 0.764 | 0.884 | 0.843 | 0.791 | 0.869 | 0.850 |
| | wine | 0.619 | 0.623 | 0.605 | 0.605 | 0.616 | 0.630 | 0.584 | 0.607 | 0.620 | 0.616 |
| | digits | 0.904 | 0.883 | 0.891 | 0.891 | 0.729 | 0.811 | 0.737 | 0.562 | 0.755 | 0.851 |
| | cancer | 0.812 | 0.823 | 0.797 | 0.797 | 0.764 | 0.703 | 0.798 | 0.804 | 0.829 | 0.783 |
| | faces | 0.640 | 0.613 | 0.618 | 0.621 | 0.538 | 0.601 | 0.566 | 0.502 | 0.623 | 0.614 |
| Dasgupta | iris | 320665 | 320883 | 320665 | 320665 | 362177 | 322308 | 314445 | 323384 | 310957 | 311267 |
| | wine | 29114 | 29093 | 29114 | 29114 | 29468 | 27095 | 27891 | 27745 | 27324 | 26983 |
| | digits | 243841216 | 243166244 | 243840641 | 243837090 | 244130381 | 244983907 | 244836138 | 243726701 | 240476750 | 241239871 |
| | cancer | 789808 | 751107 | 789808 | 789808 | 794858 | 952966 | 841428 | 742226 | 737071 | 752549 |
| | faces | 4629934 | 4632156 | 4629934 | 4622371 | 4674997 | 4669787 | 4640884 | 4600388 | 4569916 | 4619691 |

Table 3: Variability in each cost measure when running ParHAC using varying values of $\epsilon$ on the Iris dataset with $k = 10$. Each entry is the standard deviation of the metric when running 100 repeated trials.

| Epsilon | ARI | NMI | Purity | Dasgupta |
|---|---|---|---|---|
| 0 | 2e−16 | 3e−16 | 2e−16 | 5e−11 |
| 0.01 | 1e−16 | 3e−16 | 2e−16 | 4e−16 |
| 0.1 | 0.058 | 0.044 | 0.024 | 633 |
| 1.0 | 0.063 | 0.048 | 0.031 | 1592 |

- Figures 27–30 show quality measures for the *faces* dataset.
- Figures 31–34 show quality measures for the *cancer* dataset.

There are several interesting trends that we observe across all of these results:

- First, *sparse similarity graphs*, i.e. graphs constructed using very small $k$ relative to the total number of data points yield high quality results. In particular, using $k = 10$ is almost always an ideal choice for all quality measures except for Dasgupta cost. For the unsupervised Dasgupta cost objective, we observe that the cost actually improves slightly using very dense (almost complete) graphs, which could be due to the fact that the Dasgupta cost measures is computed over the *complete* version of the graph.

- Second, we observe that Affinity and SCC yield significantly more noisy results compared with the ParHAC results (note that Affinity and SCC are deterministic algorithms; by noise we are referring to the clustering quality as a function of $k$). It would be interesting to better understand why this is the case. We conjecture that this is due to overmerging within each round, and plan to investigate this further in future work.

- Third, ParHAC using $\epsilon = \{0, 0.01, 0.1\}$ yield similar results in terms of quality up until very large $k$. For $\epsilon = 1$, we see a sharp divergence and loss of quality, which suggests that this value of $\epsilon$ may be impractical to use in practice. We find that our suggested choice of $\epsilon = 0.1$ yields almost exact quality results for very sparse graphs (see Table 2 for the results with $k = 10$) and degrades gradually, with poor performance only once the graph becomes close to complete.

**Variability for $k = 10$.** As ParHAC is a randomized algorithm, the clustering results obtained by the algorithm can vary from run to run. We studied the variability in the clustering outputs obtained by ParHAC for different values of $\epsilon$, and for a fixed $k = 10$ in the graph-building process and show the standard deviation in Table 3. We observe that for $\epsilon = 0$ and very small values, e.g., $\epsilon = 0.01$, the clustering outputs are essentially identical across runs. For $\epsilon = 0.1$, we observe some variability across all cost measures, but note that the difference is at most $5.8\%$ for the ARI, NMI, and Purity metrics, and is just $0.19\%$ of the overall Dasgupta cost for the Dasgupta objective. We also emphasize that the higher variability seems to be responsible for some of the best results that we observe in Table 2, e.g., ParHAC using $\epsilon = 0.1$ achieves the best ARI score across all algorithms that we evaluated. When going from $\epsilon = 0.1$ to $\epsilon = 1$, as expected, we observe that the variability increases with increasing $\epsilon$, as the algorithm has greater choice in which edges to merge in a given layer.

### D.2 Scalability Evaluation

In this sub-section, we present additional experimental results on the scalability of our algorithms, both on real-world graph datasets and real-world pointsets (both described earlier).

**Speedup Results.** In Figure 6 we show the speedup of our ParHAC implementation on the RD, LJ, and OK graphs. On LJ and OK, our ParHAC implementation achieves 48.0x and 61.3x self-relative speedup respectively. For the RD graph, our ParHAC implementation achieves 25.7x self-relative speedup. The lower self-relative speedups for the RD graph are since ParHAC performs significantly less work on RD than on the LJ graph. In particular, the time spent merging vertices in the MultiMerge implementation is 3.8x lower than the time spent in the LJ graph, although both graphs have nearly the same total number of vertices and edges. The reason is that the average number of neighbors per cluster at the time of its merge is significantly lower on RD than on the other graphs.

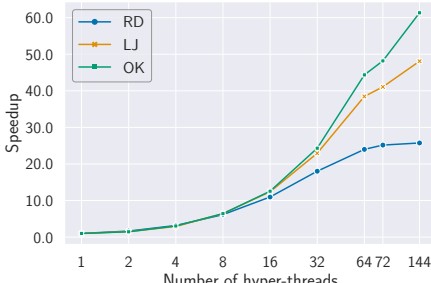

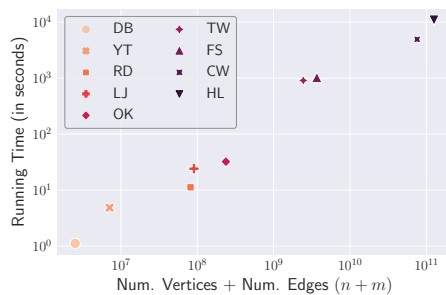

Figure 6: Speedups for three of our large real-world graphs on the $y$-axis versus the number of hyper-threads used on the $x$-axis.

Figure 7: Parallel running time of the ParHAC algorithm in seconds on the $y$-axis in log-scale versus the size of each graph in terms of the total number of vertices and edges on the $x$-axis.

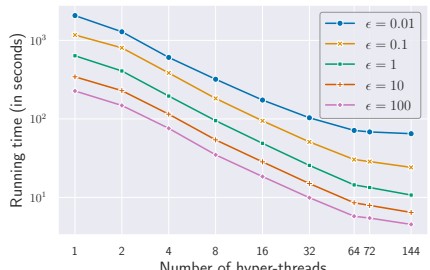

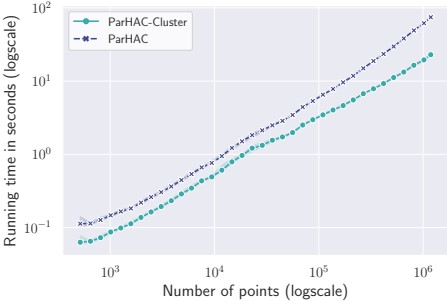

Figure 8: Parallel running times of the ParHAC algorithm on the LJ graph in log-scale as a function of the number of threads for varying values of the accuracy parameter, $\epsilon$.

Figure 9: Variability in the end-to-end running times of ParHAC using $\epsilon = 0.1$ and 144 hyper-threads on varying-size slices of the glove-100 dataset. The error lines show the 95% confidence interval for 10 independent runs per point. The running times shown for ParHAC include the cost of computing approximate $k$-NN and generating the input similarity graph. ParHAC-Cluster reports just the time taken to cluster the generated similarity graph.

**Scalability with Increasing Graph Sizes.** Figure 7 shows the parallel running times of ParHAC as a function of the graph size in terms of the total number of vertices and edges in the graph. We observe that the running time of our algorithms grows essentially linearly as a function of the graph size. We noticed that although the total number of vertices and edges for the RD and LJ graphs are similar (LJ has 10% more total vertices and edges), the running time for the LJ graph is 33% larger. The reason is that the RD graph has significantly lower average-degree than the LJ graph, which results in much less work being performed on average when merging vertices.

**Scalability for Varying Epsilon.** In the next experiment, we study the absolute performance and speedup achieved by our ParHAC implementation on a fixed graph, as $\epsilon$ is varied. Figure 8 shows the results on the LJ graph for $\epsilon \in \{0.01, 0.1, 1, 10, 100\}$. We observe that larger $\epsilon$ consistently results in lower running times, and that the value of $\epsilon = 0.1$ which we use in our quality and other scalability experiments requires the second highest running times. We observed similar results on our other graph inputs. On 144 hyper-threads using a value of $\epsilon = 1$ provides a 2.25x speedup, and $\epsilon = 100$ yields a 5.32x speedup over the parallel running time of $\epsilon = 0.1$.

**Performance Variability in End-to-End Experiments.** Figure 9 shows the performance variability of our end-to-end experiment from Section 3 (Figure 2). We observe that the performance variability across multiple runs is extremely small, with the exception of the first few data points, which

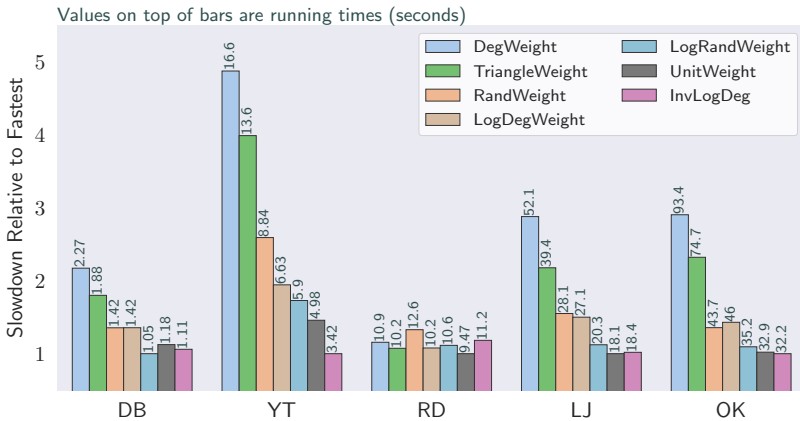

Figure 10: Effect of different weight schemes on the parallel 144-thread running time of ParHAC using $\epsilon = 0.1$.

experience slightly more variability due to scheduling variability when running on very small datasets. Over all slices of the dataset that we evaluate, the largest ratios between the slowest and fastest times for a point is at most $1.31\times$ for the time to run ParHAC, and $1.17\times$ for the end-to-end time, and considering only inputs with $n > 30000$ points, the largest ratios are $9.34\times$ and $4.1\times$ respectively.

**Different Weight Schemes.** To understand the effect of weights on the performance of ParHAC, we generated five different versions of the RD, LJ, and OK graphs using different weighting schemes where we set the weight of a $(u, v)$ edge as follows:

(1) *DegWeight*: $w(u, v) = deg(u) + deg(v)$
(2) *TriangleWeight*: $w(u, v) = |N(u) \cup N(v)|$
(3) *RandWeight*: $w(u, v) = \mathsf{Uniform}[1, 2^{32}]$
(4) *LogDegWeight*: $w(u, v) = \log(deg(u) + deg(v))$
(5) *LogRandWeight*: $w(u, v) = \log(\mathsf{Uniform}[1, 2^{32}])$
(6) *UnitWeight*: $w(u, v) = 1$
(7) *InvLogDeg*: $w(u, v) = 1/\log(deg(u) + deg(v))$

Note that LogInvWeight is the default weighting scheme used when weighting our unweighted graph inputs.

Figure 10 shows the results of the experiment We found that different weighting schemes has an impact on the running time, but only up to a small constant factor (this is despite the very large difference in the aspect ratios across the different weighting schemes being in some cases). Different weight schemes, especially those based on Degree such as *DegWeight* and *TriangleWeight* require larger overheads on power-law degree distributed graphs like YT, LJ, and OK. In contrast, on RD, all the schemes perform essentially the same, which is due to the fact that the maximum and average-degrees on this graph are a small constant, and since there are very few triangles incident to each edge. Furthermore, we observe that across all graphs, even costly schemes such as *DegWeight* which encourage high-degree vertices to merge with each other can be accurately solved with at most a 5x overhead over schemes like *Unit* or *InvLogDeg*.

**Comparison with Other Algorithms.** Figure 3 shows the relative performance and parallel running times of ParHAC compared with the SeqHAC and SeqHAC$_\mathcal{E}$ algorithms, and our implementations of the RAC, Affinity, and SCC$_\mathrm{sim}$ algorithms. We prefix the names of new implementations (where no shared-memory parallel implementation was previously available) with Par.

We observe that our Affinity implementation is always the fastest on our graph inputs. The reason is that on average Affinity requires just 8.7 iterations to complete on these inputs. On the other hand, SCC$_\mathrm{sim}$, which runs Affinity with weight thresholding, uses all 100 iterations, and is an average of 11.5x slower than Affinity due to the cost of the additional iterations. Our results show that when carefully implemented, Affinity is a highly scalable algorithms that can cluster graphs with billions of edges in a matter of just tens of minutes, although unlike ParHAC and the other approximate and exact HAC algorithms, they do not provide strong theoretical guarantees.

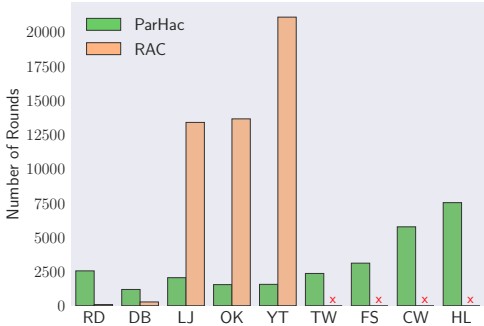

Figure 11: Rounds required by the ParHAC (using $\epsilon = 0.1$) and reciprocal agglomerative clustering (RAC) algorithms for six large real-world graphs. The ClueWeb (CW) and Hyperlink (HL) graphs are two of the largest publicly available graphs, with 978M and 1.72B vertices, and 74B and 124B edges, respectively. We mark experiments where RAC does not finish after 6 hours with a red x.

Compared to Affinity and $SCC_{sim}$, ParHAC is an average of 14.8x slower than our Affinity implementation and 1.24x slower than our $SCC_{sim}$ implementation. The main reason for the relative speed of ParHAC compared with Affinity is the much larger number of rounds required by ParHAC on our graph inputs (see Figure 11). We note that running $SCC_{sim}$ with fewer rounds yields faster results, but used 100 iterations since this setting yielded the highest quality results in our evaluation in Section 3.1. Furthermore, for all of our graph inputs, Affinity has one (or a few) rounds where the graph shrinks by a massive amount, suggesting the formation of giant cluster(s) through chaining, a known issue with clustering methods based on Boruvka's algorithm [53]. For example, the first round of Affinity on the CW graph drops the number of edges from 74.7B to 5.31B (14x lower) and the number of vertices from 955M to 118M. Importantly, the very first round forms a cluster containing 261M vertices. To conclude, our results show that ParHAC achieves a good compromise between running time and quality, as our study in Section 3.1 shows that both Affinity and $SCC_{sim}$ can produce sub-optimal dendrograms compared to the greedy exact HAC baseline.

Compared with other methods that provide provable guarantees compared with HAC, ParHAC is significantly faster. Compared with SeqHAC and SeqHAC$_{\mathcal{E}}$, ParHAC obtains 45.2x and 72.7x speedup on average respectively. Compared with RAC, ParHAC achieves an average speedup of 7.1x. Although RAC is faster than ParHAC on DB and RD, two of our small graph inputs, it seems to require a very large number of rounds on the remaining graphs, as we show in Figure 11. In particular, it can take a linear number of steps in the worst case [73], and, as we show in Figure 11, up to 21,081 steps on the YouTube (YT) real-world graph with just 1.1M vertices and 5.9M edges, and an even larger number of rounds on our larger datasets, where RAC does not terminate within 6 hours. Since each round computes the best edge for each vertex for a total of approximately $O(m)$ work, and the number of rounds for our larger graphs is close to $O(\sqrt{m})$ based on Figure 11, the super-linear total work of this algorithm prevents it from achieving good scalability. Similarly, we note that ParHAC$_{\mathcal{E}}$ (i.e. using $\epsilon = 0$) only runs within the time limit for DB, YT, and RD, and therefore it is not shown in Figure 3. The reason is that it performs $O(mn)$ work when $\epsilon = 0$, since each iteration costs $O(m)$ work and the number of outer-rounds is $O(n)$.

**Discussion.** To the best of our knowledge, our results are the first to show that graphs with tens to hundreds of billions of edges can be clustered in a matter of tens of minutes (using heuristic methods like Affinity and $SCC_{sim}$ with fewer iterations) to hours (using $SCC_{sim}$ with many iterations or methods with approximation guarantees such as ParHAC). We are not aware of other shared-memory clustering results that work at this large scale. Our theoretically-efficient implementations can be viewed as part of a line of work showing that theoretically-efficient shared-memory parallel graph algorithms can scale to the largest publicly available graphs using a modest amount of resources [31–33].

### D.3 Ablation: Comparing CPAM with Hash-based Representations

When designing our implementation of ParHAC, and specifically the clustered graph representation, we also considered other implementations of a clustered graph. Specifically, we considered other ways of representing the out-neighborhood of a vertex (cluster). To better understand the impact of using

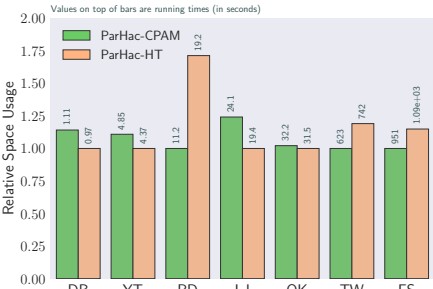

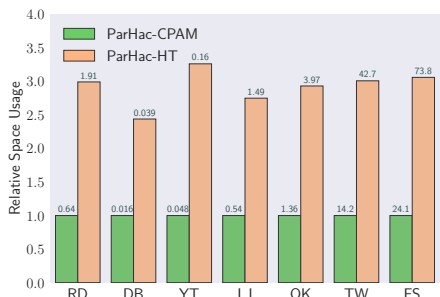

Figure 12: Relative running times of ParHAC-CPAM (cluster neighborhoods represented using compressed purely-functional trees from the CPAM framework) and ParHAC-HT (cluster neighborhoods represented using parallel hashtable). The values shown on top of each bar are the running time in seconds using 144 hyper-threads.

Figure 13: Relative sizes of the memory representation in bytes of ParHAC-CPAM (cluster neighborhoods represented using compressed purely-functional trees from the CPAM framework) and ParHAC-HT (cluster neighborhoods represented using parallel hashtables). Weights are stored as uncompressed floats. The values shown on top of each bar are the space usage of each representation in gigabytes (GiB).

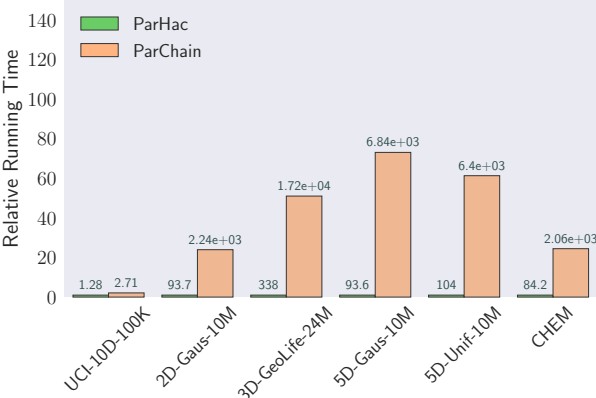

Figure 14: Performance of ParHAC compared with ParChain on the large low-dimensional pointset inputs used in the ParChain paper. The running times for ParHAC include the similarity graph construction time.

CPAM to represent this weighted adjacency information, we implemented a new *hashtable-based* implementation of the clustered graph object where each cluster stores its neighbors in a hashtable (table). Our hashtable representation uses the linear-probing concurrent hashtable described by Shun et al. [68] to represent the weights. In our implementation of a hash-based clustered graph, each hashtable is keyed by the id of the neighbor and the value stored is the weight of the edge. We call the CPAM-based implementation ParHAC-CPAM and the hashtable-based implementation ParHAC-HT.

To understand the impact of using CPAM, we studied the parallel running time of our algorithms using ParHAC-CPAM and ParHAC-HT, as well as the space usage of both types of representations. Figure 12 and Figure 13 show the relative running times and relative sizes, respectively. We find that the two implementations achieve very similar running times across all graphs (the one notable exception is the RD graph, which has very low average degree, and therefore results suffers some overhead due to hashing in the hashtable-based implementation). On the other hand, the space usage of ParHAC-CPAM is consistently much better than that of ParHAC-HT, with an average space improvement of 2.9x.

### D.4 Comparing ParHAC with ParChain

In addition to our comparison with Fastcluster in the main text of the paper, we wanted to understand how ParHAC compares with state-of-the-art multicore parallel metric HAC implementations. We

selected ParChain [79], since this recent algorithm demonstrates performance improvements over other existing parallel implementations of metric HAC.

To understand how ParHAC and ParChain compare, we ran ParHAC on a subset of the largest pointset datasets used in the ParChain paper [79], using the same end-to-end approach described earlier (graph building using an implementation of Vamana, followed by running ParHAC). Figure 14 shows the results of the experiment. We find that on average, ParHAC is 39.3x faster, and up to 73x faster than ParChain for the average-linkage measure. We note that when we tried to run ParChain on the same high-dimensional inputs used in this paper, e.g., Glove-100, the algorithm crashed, which is likely due to the fact that ParChain and its optimizations were designed for low-dimensional (approximately $d \leq 16$) HAC. We also ran ParChain on the UCI datasets, and found that it produced identical dendrograms to those computed by Sklearn, FastCluster, and SciPy, confirming that ParChain is an exact algorithm (and is implemented correctly).

The datasets used in our experiments can be found in the ParChain paper, and are also described below for completeness:

**ParChain Datasets.** The *GaussianDisc* data set contains points inside a bounding hypergrid with side length $5\sqrt{n}$, where $n$ is the total number of points. $90\%$ of the points are equally divided among five clusters, each with a Gaussian distribution. Each cluster has its mean randomly sampled from the hypergrid, a standard deviation of $1/6$, and a diameter of $\sqrt{n}$. The remaining points are randomly distributed. The *UniformFill* data set contains points distributed uniformly at random inside a bounding hypergrid with side length $\sqrt{n}$, where $n$ is the total number of points. These datasets are synthetically generated, and thus no license is applicable to the best of our knowledge. The synthetic data sets contain 10 million points for dimensions $d = 2$ and $d = 5$. Lastly, *UCI1* [3, 16] is a 10-dimensional data set with 19020 data points.

*GeoLife* [2, 80] is a 3-dimensional real-world data set with 24876978 points. This dataset contains user location data, and is extremely skewed. The data is provided as part of a Microsoft Research project, and no license information is available online to the best of our knowledge. *CHEM* [1, 40] is a 16-dimensional dataset with 4208261 points containing chemical sensor data. Similar to the other datasets from UCI, no licensing information is available, although the project states that the dataset has been donated to UCI by the creators of the dataset (from UCSD).

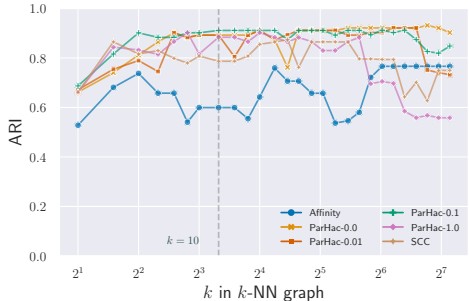

Figure 15: Adjusted Rand-Index (ARI) of clusterings computed by ParHAC for varying $\epsilon$ on Iris versus the $k$ used in similarity graph construction.

Figure 16: Normalized Mutual Information (NMI) of clusterings computed by ParHAC for varying $\epsilon$ on Iris versus the $k$ used in similarity graph construction.

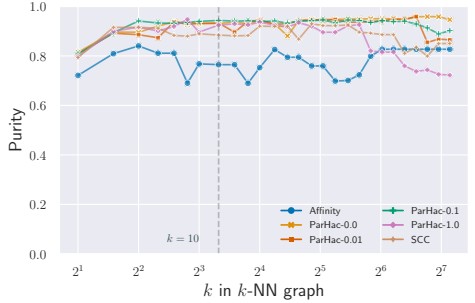

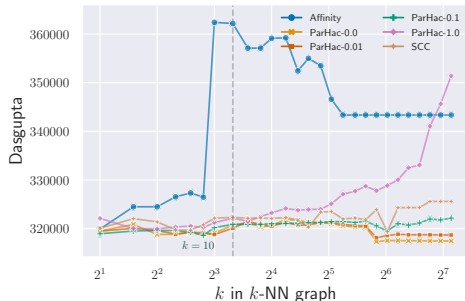

Figure 17: Dendrogram Purity (Purity) of clusterings computed by ParHAC for varying $\epsilon$ on Iris versus the $k$ used in similarity graph construction.

Figure 18: Dasgupta Cost (Dasgupta) of clusterings computed by ParHAC for varying $\epsilon$ on Iris versus the $k$ used in similarity graph construction.

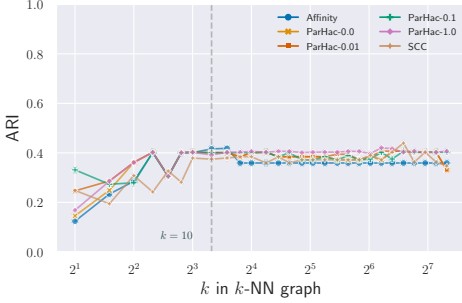

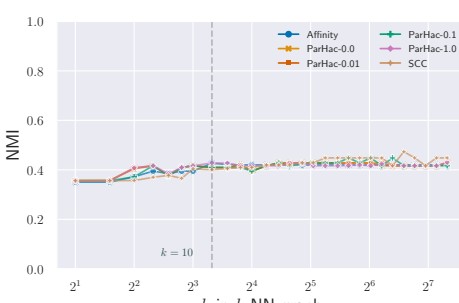

Figure 19: Adjusted Rand-Index (ARI) of clusterings computed by ParHAC for varying $\epsilon$ on Wine versus the $k$ used in similarity graph construction.

Figure 20: Normalized Mutual Information (NMI) of clusterings computed by ParHAC for varying $\epsilon$ on Wine versus the $k$ used in similarity graph construction.

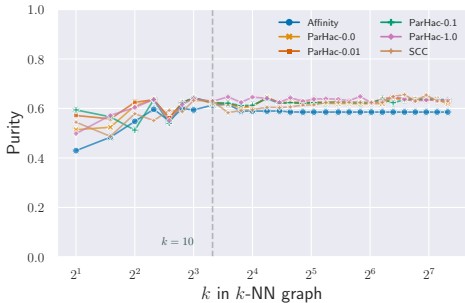

Figure 21: Dendrogram Purity (Purity) of clusterings computed by ParHAC for varying $\epsilon$ on Wine versus the $k$ used in similarity graph construction.

Figure 22: Dasgupta Cost (Dasgupta) of clusterings computed by ParHAC for varying $\epsilon$ on Wine versus the $k$ used in similarity graph construction.

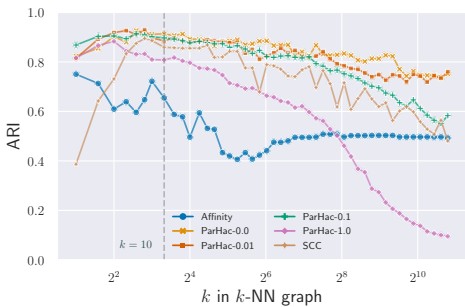

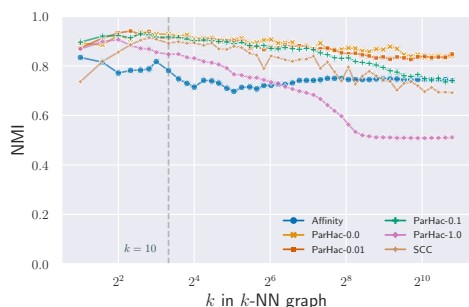

Figure 23: Adjusted Rand-Index (ARI) of clusterings computed by ParHAC for varying $\epsilon$ on Digits versus the $k$ used in similarity graph construction.

Figure 24: Normalized Mutual Information (NMI) of clusterings computed by ParHAC for varying $\epsilon$ on Digits versus the $k$ used in similarity graph construction.

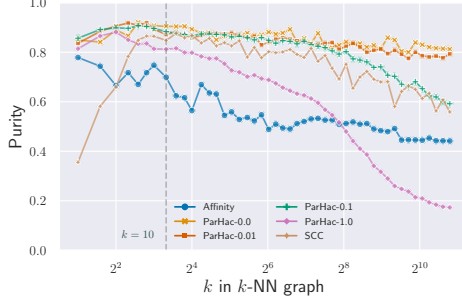

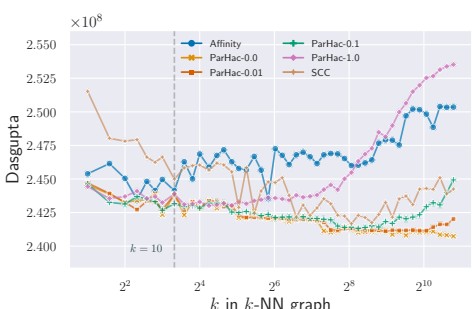

Figure 25: Dendrogram Purity (Purity) of clusterings computed by ParHAC for varying $\epsilon$ on Digits versus the $k$ used in similarity graph construction.

Figure 26: Dasgupta Cost (Dasgupta) of clusterings computed by ParHAC for varying $\epsilon$ on Digits versus the $k$ used in similarity graph construction.

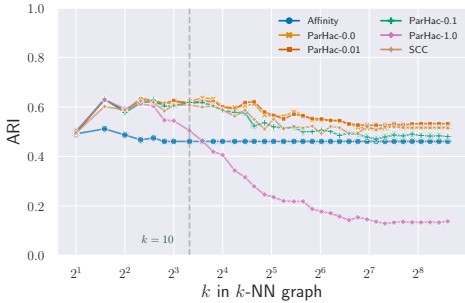

Figure 27: Adjusted Rand-Index (ARI) of clusterings computed by ParHAC for varying $\epsilon$ on Faces versus the $k$ used in similarity graph construction.

Figure 28: Normalized Mutual Information (NMI) of clusterings computed by ParHAC for varying $\epsilon$ on Faces versus the $k$ used in similarity graph construction.

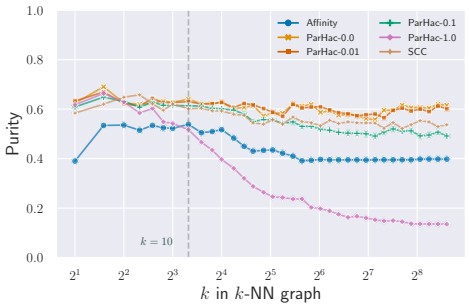

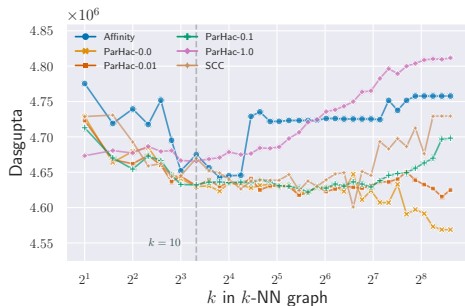

Figure 29: Dendrogram Purity (Purity) of clusterings computed by ParHAC for varying $\epsilon$ on Faces versus the $k$ used in similarity graph construction.

Figure 30: Dasgupta Cost (Dasgupta) of clusterings computed by ParHAC for varying $\epsilon$ on Faces versus the $k$ used in similarity graph construction.

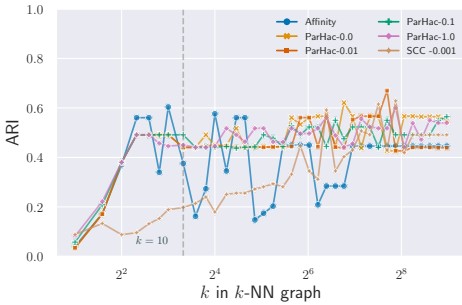

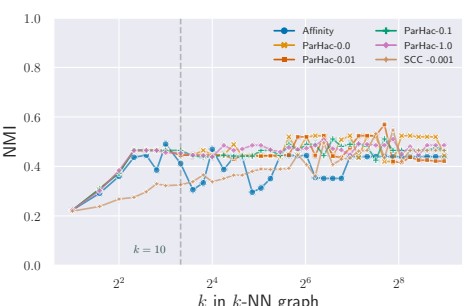

Figure 31: Adjusted Rand-Index (ARI) of clusterings computed by ParHAC for varying $\epsilon$ on Cancer versus the $k$ used in similarity graph construction.

Figure 32: Normalized Mutual Information (NMI) of clusterings computed by ParHAC for varying $\epsilon$ on Cancer versus the $k$ used in similarity graph construction.

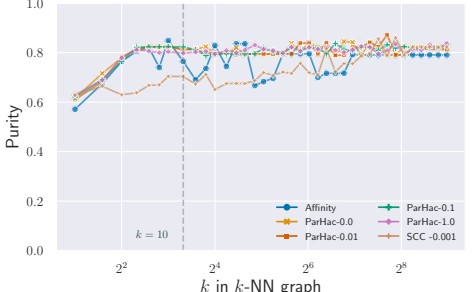

Figure 33: Dendrogram Purity (Purity) of clusterings computed by ParHAC for varying $\epsilon$ on Cancer versus the $k$ used in similarity graph construction.

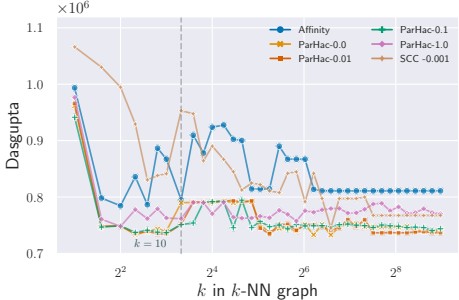

Figure 34: Dasgupta Cost (Dasgupta) of clusterings computed by ParHAC for varying $\epsilon$ on Cancer versus the $k$ used in similarity graph construction.