# OpenReview forum: "Hierarchical Agglomerative Graph Clustering in Poly-Logarithmic Depth "
_NeurIPS.cc/2022/Conference — NeurIPS 2022 Accept_

### Official Review · Reviewer_zNhm · 2022-06-20

**Rating:** 5
**Confidence:** 3
**Soundness:** 3 good
**Presentation:** 2 fair
**Contribution:** 3 good

**Summary:**

This paper presents an efficient parallel hierarchical agglomerative clustering (HAC) algorithm with sublinear depth for the average linkage function. In particular, the authors provide a (1+$\epsilon$)-approximation algorithm on $m$ edge graphs using $\tilde{O}(m)$ time and poly-logarithmic depth. Experiments show that the proposed ParHAC algorithm enjoys huge speedup compared with state-of-the-art sequential and parallel baselines while achieving similar quality.

** Post Rebuttal Update **
I've read the rebuttal and the other reviewers' comments. I appreciate the update the authors have made, and I have no further questions given the inputs.

**Questions:**

Q1. For the pointset experiment, one might want to discuss the methods of generating the pairwise similarity.

Q2. Since the experiments include a variety of datasets, the authors might want to summarize and compare the datasets.

**Limitations:**

No conclusion. No discussion on potential societal impact.

**Strengths And Weaknesses:**

** Strengths **

S1. The theoretical analysis provides a good understanding of the algorithm design's general idea (Thm 1.1, 1.2), the hardness (lower bound), and the performance (Thm 2.3).

S2. Experiments show substantial speedup over the baselines.

** Weaknesses **

W1. The presentation of algorithm design is unclear.

W2. The presentation of experiments is unclear. The authors might want to put more tables in the main text.

W3. The presentation seems to focus on speed and discuss less on quality. Usually one might expect a trade-off between speed and quality. Thus, discussing both should be of interest.

---

> ### Author Response · Authors · 2022-08-02
> **Response to Reviewer zNhm**
>
> > The presentation of algorithm design / experiments is unclear.
>
> Thank you for providing feedback on the presentation. We would be happy to improve the presentation, and in particular try to put more tables in the main text. However, we would appreciate a bit more specific pointers on how to improve the presentation.
>
> > For the pointset experiment, one might want to discuss the methods of generating the pairwise similarity.
>
> We generated the similarities used in our pointset experiments using the formula $\mathsf{sim}(u,v) = 1/(1 + \mathsf{dist}(u,v))$. We then reweighted the similarities so that the similarities lie in [0, 1] by dividing by the maximum similarity (this is briefly described at the start of Section 3). Regarding why we chose this method, we acknowledge that there are many ways of converting distances to similarities. In fact, we previously tried other conversion functions and actually obtained better quality results using these functions, e.g., $\mathsf{sim}(u,v) = 1/(1+\log^3(\mathsf{dist}(u,v))$, but chose the final formula above since (a) this similarity function has been previously used in the literature and (b) the relative ranking of results did not seem to depend on the similarity function, and (c) tuning the similarity function seemed like a separate topic, and would have distracted from the main focus of the paper. We will add a brief comment with respect to our choice of similarity function to the paper describing our rationale.
>
> > Since the experiments include a variety of datasets, the authors might want to summarize and compare the datasets.
>
> Thanks for this helpful suggestion. We described the graph datasets used in the supplementary materials (in Section D), and will put a pointer to them in the main text. We will also add a description of the UCI datasets used, and a discussion and comparison of them, for example discussing the ground truth clusters, to Section D.

---

### Official Review · Reviewer_vzbQ · 2022-07-08

**Rating:** 7
**Confidence:** 3
**Soundness:** 3 good
**Presentation:** 4 excellent
**Contribution:** 3 good

**Summary:**

This paper focuses on a parallel approximation algorithm for hierarchical agglomerative clustering (HAC) with average—linkage function. Given $n$ data points, it constructs a similarity graph by considering each point as a vertex and connecting it to its $k$ nearest neighbors. The algorithm then separates the graph into logarithmic geometric layers such that each layer only contains edges of the same weight (up to 1+eps factor). Within each layer, all edges with weight within 1+eps factor of the maximum weight called heavy edges are merged to a larger cluster in parallel. Here a smart randomized algorithm is designed to merge all heavy edges. It can be proved that the algorithm runs in $\tilde{O}(m+n)$ work, and more importantly, in polylogarithmic depth with high probability. Finally, extensive experiments have been performed in graphs of different scales, to answer questions including: what is the clustering quality of the approximate algorithm compared to the exact HAC, HAC with complete similarity matrix, and existing works; how to tune the approximate parameter eps and similarity graph parameter $k$ for high quality results in practice; what is the run-time performance compared with the baselines mentioned above. Overall, the proposed algorithm empirically achieves a good balance between clustering quality and running time, and can scale to graphs of hundreds of billions edges.

**Questions:**

See Weaknesses items 2 and 3.

**Strengths And Weaknesses:**

Strengths:
1. Develops a simple yet efficient parallel algorithm for approximating HAC with average-linkage function. The poly-logarithmic depth is proposed for the first time, although it is based on a standard geometric layering for converting weighted graphs to unweighted graphs (within 1+eps factor).
2. Conduct comprehensive experimental evaluation and compare against recent baselines, exact counterpart, HAC based on complete similarity matrix, and end-to-end pointset-based clustering. Compared with the exact counterparts, this approximate algorithm achieves a clustering quality close to that of the exact algorithm across different metrics. The parallel algorithm provides an order of magnitude speedup over the sequential algorithm it is based on.
3. The presentation of this work is good. The main text well summarizes the high-level ideas of their algorithm, with sufficient details included in the Appendix. Similarly, the experimental parts demonstrate the benefits of their algorithm, analyze the efficiency aspect, and refer to complete results in the Appendix.

Weaknesses:
1. The computational cost in terms of running time required by the current algorithm is an order of magnitude worse than Affinity, ~1.2 times worse than SCC on average. As in Figure 1, the clustering quality of SCC are comparable to ParHAC for a small value of $k$, which is the best parameter range one will choose.
2. It is kind of counter-intuitive to see that a larger value of $k$ in building similarity graphs results in worse clustering quality. But, when the complete similarity graphs are used as in SciPy package (k=n), the performance is not too bad. It is unknown why this phenomenon is observed across the implementations of multiple algorithms.
3. Existing works that can scale to trillion-edge graphs have been reported such as SCC. ParHAC cannot achieve that scale because of memory constraints. Does this mean that ParHAC is an in-memory algorithm? Then what is the challenge when adapting it to an on-disk algorithm that poses less requirements on the memory?

---

> ### Author Response · Authors · 2022-08-02
> **Response to Reviewer vzbQ**
>
> > Regarding “[the algorithm] is based on a standard geometric layering”
>
> We would like to emphasize that while the starting point of our algorithm is geometric layering, our main algorithmic contribution is the parallel algorithm which deals with each layer (Algorithm 1) and its theoretical analysis. We discuss the challenges in designing the algorithm in section A.2 in supplementary material, and provide a complete analysis in section A.3.
>
> We will also expand the discussion of the challenges involved in designing the algorithm in the main body of the paper.
>
> > “It is kind of counter-intuitive to see that a larger value of $k$ in building similarity graphs results in worse clustering quality. But, when the complete similarity graphs are used as in SciPy package ($k=n$), the performance is not too bad. It is unknown why this phenomenon is observed across the implementations of multiple algorithms.”
>
> Our experience is that using a sparse $k$-NN graph before running a clustering algorithm can help eliminate noise from the input. For instance, studying the fraction of the $k$-NN graph edges that are within-cluster vs. between-cluster (with respect to the ground truth clustering), we see a sharp decline in the amount of within-cluster edges as we increase $k$. For instance, on the digits dataset, we obtain the following fractions of intra-cluster and inter-cluster edges
>
> $k$  | fraction of intra-cluster edges | fraction of inter-cluster edges
> ---- | ---------------------------- | -------------------------------
> 10   | 0.81                         | 0.19
> 50   | 0.72                         | 0.28
> 100  | 0.61                         | 0.39
> 160  | 0.52                         | 0.47
> 1774 | 0.1                          | 0.9
>
>
> Similar trends hold in the other datasets. Since all of the clustering algorithms work on the same underlying similarity graphs, we suspect that the fact that the algorithm results degrade in quality as $k$ increases follows from the gradual introduction of noisy between-cluster edges in the underlying graph.
>
> We also observe, as seen in Figure 1, that higher values of epsilon suffer more in the large-$k$ setting, and that for $\epsilon=0$, our exact algorithm achieves a similar level of performance as the SciPy implementation on the complete graph (the small difference is likely due to SciPy using dissimilarities and our implementations using similarities). We agree with the reviewer that this phenomenon is counter-intuitive, and plan to add more empirical data illustrating this effect to the supplementary materials.
>
>
> > Existing works that can scale to trillion-edge graphs have been reported such as SCC. ParHAC cannot achieve that scale because of memory constraints. Does this mean that ParHAC is an in-memory algorithm? Then what is the challenge when adapting it to an on-disk algorithm that poses less requirements on the memory?
>
> This is correct. ParHAC was designed with the shared-memory parallel setting in mind, as opposed to e.g. SCC which was designed for the distributed setting using multiple machines. While the number of iterations executed by ParHAC is small enough to achieve good parallelism, it can still be relatively large (e.g. about 8000 iterations on the 100B-edge HyperLink graph). If we implemented ParHAC using external storage (e.g. disk or a high-performance distributed storage) this number of iterations would likely become a bottleneck, as in the current design most of the graph would need to be read in each iteration. It is an interesting question whether one can give a HAC algorithm executing fewer rounds, which would open the way to distributed implementations and implementations using external storage.

---

### Official Review · Reviewer_w4sm · 2022-07-10

**Rating:** 6
**Confidence:** 3
**Soundness:** 3 good
**Presentation:** 3 good
**Contribution:** 2 fair

**Summary:**

This paper develops a parallel hierarchical agglomerative clustering (HAC) algorithm ParHAC which achieves $(1+\epsilon)$-approximation with $\tilde{O}(m)$ work and poly-log depth. It also shows that the graph-based HAC using average-linkage is P-complete. The experiments on real datasets verifies the accuracy and scalability of ParHAC.

**Questions:**

(1) What is the relationship of this work with [23]?

(2) The authors explain the "work" as the number of operations performed in Sec 1.1. What kinds of operations does it count? Are they MultiMerge and Neighborhood Primitives? Why do the authors use work rather than running time in Th 1.1? What is the running time of the algorithm?

**Limitations:**

I cannot find any issue with negative societal impact. The main concerns have been listed in questions.

**Strengths And Weaknesses:**

The speedup and scalability of ParHAC are significant, and the P-completeness result is new. The key idea of layering, which is a common proof skill in approximation algorithms, is novel for the proof of $(1+\epsilon)$-approximation, but it seems to be enlightened by [23]. The result of $(1+\epsilon)$-approximation and its proof seem similar to those in [23] also, except for the poly-log depth. It is better for the authors to make a thorough comparison with [23] and clarify the difference.

---

> ### Author Response · Authors · 2022-08-02
> **Response to Reviewer w4sm**
>
> > (1) Relationship of this work with [23]
>
> [23] showed an algorithm for $(1+\epsilon)$-approximate HAC, which runs in near-linear time. Translating this result directly into the work-depth model, this yields an algorithm with near-linear work, and near-linear depth (i.e. an algorithm which does not enjoy any speedup from adding more processors). In this paper, we show that we can reduce the depth to only polylogarithmic, while keeping the work near-linear.
>
> We believe that the main contribution of [23] in the area of approximate HAC is sending an important *conceptual* message. That is, the paper observes that considering $(1+\epsilon)$-approximate HAC leads to an algorithm with near-linear running time. However, the algorithm in [23] itself is based on a relatively straightforward observation that it suffices to update cluster-related information only when the cluster increases its size by a $(1+\epsilon)$ factor.
>
> In contrast, our theoretical contribution is *algorithmic* in nature. That is, even after applying geometric layering, the algorithm we propose for dealing with each layer (Algorithm 1) and its analysis require nontrivial insights. We also provide an efficient parallel implementation and achieve great speedups.
>
> > (2) Regarding work of operations, the work-depth model and running time
>
> Thank you for pointing out these confusions regarding the work-depth model. We will clarify these concepts in our introduction of the work-depth model.
>
> Since the running time of parallel algorithms depends on the number of processors, we use the work-depth model to describe the theoretical complexity of our algorithms; the work and depth bounds are agnostic to the number of processors, and the running time for a given number of processors can be derived from the work and depth. The work-depth model is a popularly-used model for describing shared-memory parallel algorithms (“Introduction to Algorithms (3. ed.)”. Cormen, Leiserson, Rivest, Stein. 2009; “Introduction to Parallel Algorithms”. Jaja. 1992). The work-depth model is also the primary parallel model used in a variety of recent papers on shared-memory multicore algorithms for graph problems, e.g., [20, 22], pointset clustering [46], as well as tree-based algorithms [21, 24] (as well as “PAM: Parallel Augmented Maps”. Sun et al. PPoPP’2018). Deriving good bounds for the work and depth of an algorithm also implies good bounds for the problem in a variety of classical models for parallelism such as different PRAM variants due to efficient simulation results ([8] presents an overview).
>
> The work is the total number of primitive operations (neither multimerge nor neighborhood primitives; by primitive, we mean basic computations, such as value assignment or arithmetic). The work is also equivalent to the running time of a parallel algorithm given only a single processor. The depth is the longest series of sequential dependencies, and can be colloquially thought of as the running time on an ideal computer with infinite available processors.
>
> Given a work bound of $W$ and a depth bound of $D$, Brent’s law states that the running time using $P$ processors is upper-bounded by $O(D + W / P)$. Thus, it is desirable to reduce both the work and depth in designing a parallel algorithm. The running time of our algorithm given $P$ processors is $O(\log^4(n) + m * \mathsf{polylog}(n) / P)$.

---

### Meta-Review · Area_Chair_DBUi · 2022-08-22

**Recommendation:** Accept
**Confidence:** Certain

**Metareview:**

Average score is 6. I also believe that this is a good paper. Please update the final version based on the reviewer feedback.

**Award:**

No

---

### Decision · Program_Chairs · 2022-09-14

Accept